# Mining GWAS and eQTL data for CF lung disease modifiers by gene expression imputation

Hong Dang[1]*, Deepika Polineni[2], Rhonda G. Pace[1], Jaclyn R. Stonebraker[1], Harriet Corvol[3,4], Garry R. Cutting[5,6], Mitchell L. Drumm[7], Lisa J. Strug[8,9], Wanda K. O'Neal[1], Michael R. Knowles[1]

**1** Marsico Lung Institute, University of North Carolina at Chapel Hill School of Medicine Cystic Fibrosis/ Pulmonary Research & Treatment Center, Chapel Hill, North Carolina, United States of America, **2** University of Kansas Medical Center, Kansas City, Kansas, United States of America, **3** Pediatric Pulmonary Department, Assistance Publique-Hôpitaux sde Paris (AP-HP), Hôpital Trousseau, Institut National de la Santé et la Recherche Médicale (INSERM) U938, Paris, France, **4** Sorbonne Universités, Université Pierre et Marie Curie (UPMC), Paris 6, Paris, France, **5** McKusick-Nathans Institute of Genetic Medicine, Baltimore, Maryland, United States of America, **6** Department of Pediatrics, Johns Hopkins University School of Medicine, Baltimore, Maryland, United States of America, **7** Department of Pediatrics, School of Medicine, Case Western Reserve University, Cleveland, Ohio, United States of America, **8** Department of Molecular Genetics, University of Toronto, Toronto, Ontario, Canada, **9** Division of Biostatistics, Dalla Lana School of Public Health, University of Toronto, Toronto, Ontario, Canada

* dangh@email.unc.edu

**Data Availability Statement:** All predictive models derived from GTEx human reference data set are publicly available. Gene expression data from CF LCL samples are available from GEO (accession

## Abstract

Genome wide association studies (GWAS) have identified several genomic loci with candidate modifiers of cystic fibrosis (CF) lung disease, but only a small proportion of the expected genetic contribution is accounted for at these loci. We leveraged expression data from CF cohorts, and Genotype-Tissue Expression (GTEx) reference data sets from multiple human tissues to generate predictive models, which were used to impute transcriptional regulation from genetic variance in our GWAS population. The imputed gene expression was tested for association with CF lung disease severity. By comparing and combining results from alternative approaches, we identified 379 candidate modifier genes. We delved into 52 modifier candidates that showed consensus between approaches, and 28 of them were near known GWAS loci. A number of these genes are implicated in the pathophysiology of CF lung disease (e.g., immunity, infection, inflammation, HLA pathways, glycosylation, and mucociliary clearance) and the CFTR protein biology (e.g., cytoskeleton, microtubule, mitochondrial function, lipid metabolism, endoplasmic reticulum/Golgi, and ubiquitination). Gene set enrichment results are consistent with current knowledge of CF lung disease pathogenesis. HLA Class II genes on chr6, and *CEP72*, *EXOC3*, and *TPPP* near the GWAS peak on chr5 are most consistently associated with CF lung disease severity across the tissues tested. The results help to prioritize genes in the GWAS regions, predict direction of gene expression regulation, and identify new candidate modifiers throughout the genome for potential therapeutic development.

code GSE60690). Gene expression data from CF nasal mucosal epithelial RNAseq samples are uploaded to dbGaP for controlled access for researchers who meet the criteria for access to confidential data (https://view.ncbi.nlm.nih.gov/dbgap-controlled). Data dictionaries and variable summaries are available on the dbGaP FTP site (https://ftp.ncbi.nlm.nih.gov/dbgap/studies/phs002254/phs002254.v1.p1/). The public summary-level phenotype data may be browsed at the dbGaP study report page (http://www.ncbi.nlm.nih.gov/projects/gap/cgi-bin/study.cgi?study_id=phs002254.v1.p1). The summary GWAS data from CF Gene Modifier Consortium studies and summary results of phenotype trait association testing are publicly available at GitHub (https://github.com/danghunccf/CF-GWAS-dataMiningPaper).

**Funding:** H.D. was supported by Cystic Fibrosis Foundation grant, DANG16I0. M.R.K. was supported by Cystic Fibrosis Foundation grant, KNOWLE00A0. CFF URL: https://www.cff.org/Research/Researcher-Resources/Awards-and-Grants/ The funders had no role in study design, data collection and analysis, decision to publish, or preparation of the manuscript.

**Competing interests:** The authors have declared that no competing interests exist.

## Introduction

The International Cystic Fibrosis Gene Modifier Consortium identified 5 genome-wide significant genetic loci associated with cystic fibrosis (OMIM: 219700) lung disease severity through GWAS of 6,365 CF patients, with a chr16 locus also showing significance in some analyses [1, 2]. The GWAS signals point to genes in regions that may play a role in CF lung disease pathogenesis. Heritability studies of twins and siblings estimated that at least 50% of lung disease variability is attributable to non-*CFTR* genetic modifiers [3]. The effect sizes of the identified loci as extrapolated from the beta-coefficients range from 2.5% - 4.6% predicted forced expiratory volume in one second ($FEV_1$) [1], with a combined potential effect size to explain $< 25\%$ $FEV_1$ variation. Therefore, a large proportion of genetic influences on CF lung disease severity remain undetected, in part reflecting limited statistical power of GWAS due to multiple test penalties over millions of single nucleotide polymorphisms (SNPs).

The most common scenario explaining genetic association to phenotype is through the effects of variants on gene expression [4, 5]. Studies of genetic regulation of gene expression, *i.e.*, expression Quantitative Trait Loci (eQTL), are effective strategies and "next steps" for post-GWAS investigations to understand genetic susceptibility/modification of diseases [6, 7]. The availability of reference data sets for more than 40 human tissues by the Genotype-Tissue Expression (GTEx) consortium [5] has greatly facilitated post-GWAS research. In a survey of 44 human tissues, the GTEx consortium found that most genetic regulation of gene expression is common across multiple tissues, acting through *cis*-SNPs at promoter and enhancer sites [5]. Also using the entire set of 44 GTEx tissues, as opposed to limiting analyses to 9 pilot tissues, increased the number of trait-associated variants by 5-fold for 18 complex traits [8]. In other words, genetic regulation of gene expression, or eQTL, can be informative regardless of tissue origin of the training data set [8], and can help overcome technical deficiencies, such as small sample sizes of certain tissue data, and potential biological limitations such as unsampled developmental stage and environmental and pathogenic masking of gene expression through reverse causality.

The study of eQTLs requires gene expression and genetic variation data from the same individuals, typically testing one gene-SNP pair at a time. A recent extension of eQTL analysis is the use of machine learning and predictive modeling techniques to associate multiple genetic variants, to predict gene expression [9, 10]. The PrediXcan [9] and Transcriptome-Wide Association Studies (TWAS) [10] methods utilize small training data sets (with both genotype and expression data from the same individuals), to build predictive models, where genotypes from several *cis*-SNPs are used to predict the portion of genetic regulation of expression for each gene. Once built, these models, regardless of tissue origin, can be used to impute gene expression from large GWAS studies where only genotype data are available. The implicit assumption of these approaches is that genetic regulation of gene expression is largely preserved among human population as shown by cross cohort heritability correlation [9, 10], and that eQTLs will be conserved across different tissues for most of *cis*-eQTLs [8, 9]. The resultant (imputed) gene expression can then be analyzed for association to disease phenotypes to pinpoint the genetic regulation that is relevant to the disease process. These methods can improve statistical power through interrogating SNPs associated with gene expression regulation only, thus reducing multiple test burdens. The predictive models can also suggest the direction of gene expression regulation relating to phenotype, informing the mechanism by which SNPs affect the phenotype. In addition, by interrogating multiple *cis*-SNPs at the same time, no single SNP is required to be significant, which can uncover combinatorial effects not identified otherwise [10].

Here we report the use of PrediXcan and TWAS methods to mine the CF GWAS data for genetic regulation of gene expression associated with CF lung disease severity. We use a combination of our own CF training data sets [11, 12] and reference GTEx data sets of multiple human tissues [4, 5] to generate a list of genes with evidence of association with CF lung disease severity. Leveraging the strengths of diverse approaches [9, 10], and querying multiple tissues produced 379 potential modifier candidates. From this list, 52 consensus genes met the statistical cutoff from both approaches, and 28 of these were within 1 mega-base (Mb) of significant GWAS loci. We sought indirect validation of some of these candidate CF lung disease modifier genes by examining their known functions in literature and annotation databases, and we highlight potential relevance of some of the findings to CF biology. These genes are candidates for further experimental validation.

## Methods

The overall workflow of the study is outlined in Fig 1. The cohort study design, and demographic and clinical characteristics of the CF patients used in this study have been previously described [1]. Briefly, 5 cohorts (total 6,365 CF patients) with >90% European ancestry from US, Canada, and France were recruited by the International Cystic Fibrosis Gene Modifier Consortium, and their genome-wide genetic variance were assayed using different genotyping platforms over several years. GWAS was performed as a meta-analysis of cohort/platform combinations, using the standardized quantitative lung function score, or KNoRMA (Kulich normal residual mortality adjusted) mean $FEV_1$ percentile, as phenotype trait [1, 3]. The present study also utilized gene expression data previously interrogated for association to several CF disease phenotypes, including expression data from Affymetrix exon microarrays of 753 EBV-transformed lymphoblastoid cell lines (LCLs) from CF patients [11] and RNA-sequencing from nasal mucosal epithelial biopsies from 132 CF patients [12]. These gene expression data provided training data to build predictive models using the PredictDB_Pipeline (used by PrediXcan from Im lab) for GTEx v7 release. Models for LCL gene expression available from PredictDB repository (http://predictdb.org/ from Im lab), were compared to our CF LCL models to assess the quality of our predictive models. Full details of genetic and transcriptomic datasets utilized in the modeling, and the modeling procedures are described in S1 Methods in S4 File. Additionally, GTEx models from 48 human tissues and a large data set from Depression Genes and Networks (DGN) whole blood [13] were downloaded from the PredictDB (PrediXcan) data repository [9], and TWAS [10].

Imputed SNP genotypes from the CF GWAS cohorts [14] were used as input for PrediXcan model training [9]. Compared to the imputation reported in the GWAS studies [1], the updated version here utilized a more recent release of 1000 genomes project Phase3 (v5a) haplotype data and 101 CF whole genome sequencing data as reference panels, which improved coverages at HLA and *CFTR* regions [14].

To test for association with CF lung disease severity, the quantitative score (KNoRMA) used in the prior GWAS studies was used as a standardized CF lung phenotype trait [1–3], and the imputed gene expression from each tissue was modeled as response variable to KNoRMA in a linear model, with sex and 4 genotype principle components (PCs) as covariates. Association testing of imputed gene expression, using the PrediXcan platform [9], from the CF LCLs and CF nasal epithelial biopsies, 48 GTEx tissues, and DGN whole blood (a total of 51 human tissues), were performed using robust regression [15, 16] based on 5,756 unrelated patients. The analyses were done using the Bioconductor *LIMMA* package and the robust regression utilized iterated re-weighted least squares by the *rlm* function from the R package, *MASS*. For disease phenotype association testing using predictive models trained on CF nasal epithelial

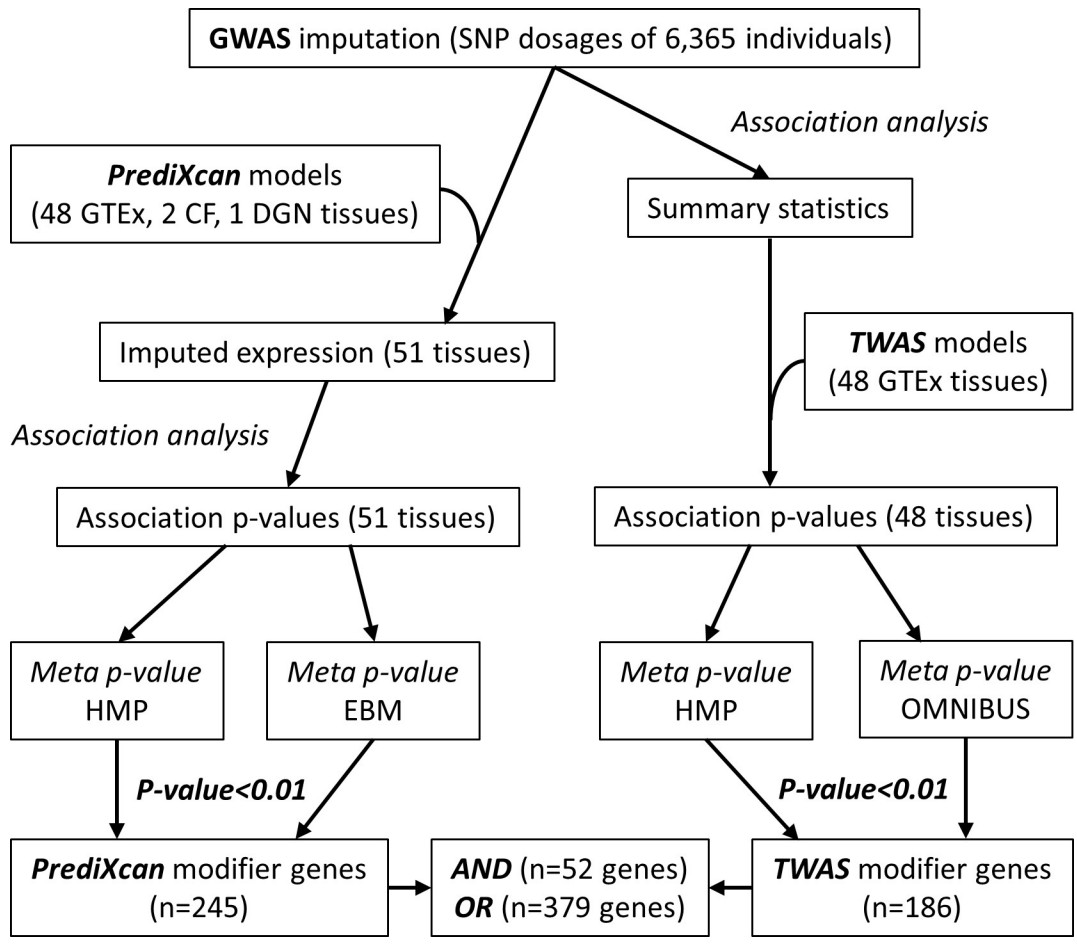

**Fig 1. Analysis workflow overview.** GWAS imputation of SNP variances in CF patients (n = 6,365) were used to impute genetically regulated gene expression, which were then tested for CF lung disease severity using either the PrediXcan platform (left arm), or TWAS (right arm). The association results from multiple tissues from each platform were combined through 2 different meta-analysis of multiple p-values from different tissues. **GTEx**: Genotype-Tissue Expression RNA-seq (n = 48 tissues); **CF**: LCL microarray (n = 753 samples), and nasal epithelial biopsy RNA-seq (n = 132 samples); **DGN**: Depression Genes and Networks RNA-seq from whole blood (n = 922 samples); **HMP**: harmonic mean p-value; **EBM**: empirical adaptation of Brown's method; **OMNIBUS**: omnibus p-value from TWAS.

biopsy and LCL data sets, the samples used in predictive model training (122 nasal and 753 LCL samples were part of GWAS) were excluded from the association testing, resulting in 5,634 and 5,003 final sample size for nasal epithelial biopsies and LCLs, respectively.

Alternatively, summary GWAS statistics were used to test imputed gene expression association from 48 GTEx tissues to KNoRMA using Functional Summary-based Imputation, or FUSION software from TWAS [10]. Briefly, summary GWAS statistics for SNP associations to CF lung disease phenotype (n = 6,365) and reference linkage-disequilibrium (LD) data from 1000 genome projects were used as input for FUSION, with TWAS predictive models from 48 GTEx v7 human tissues downloaded from FUSION website (http://gusevlab.org/projects/fusion/). The analysis was performed according to instructions on the FUSION website.

To leverage information from all tested tissues, meta-analyses from multiple p-values were performed. Since these tissue-specific association tests all started from the same CF GWAS data set, meta-analysis for dependent/correlated tests were applied to both the PrediXcan and TWAS results. We then adopted a strategy to compare results from the two independently

developed approaches. Multi-tissue tests from each result set were combined by two separate meta-analysis methods, a simple harmonic mean p-value (HMP) [17], and a correlation adjusted method, specifically, empirical adaptation of Brown's method (EBM) [18] for PrediX-can, or omnibus test [10] for TWAS. For significant modifier genes from each analysis platform, a p-value < 0.01 from both the HMP, and correlation adjusted method (EBM for PrediXcan, or omnibus for TWAS) was chosen. Consensus between the 2 result sets (with 4 p-value < 0.01 thresholds) yielded the most robust findings, while the union of significant genes from the 2 result sets maximized sensitivity of discovery. For comparison of numeric outcomes, such as performance of predictive models or imputed gene expression between data sets or tissues, the distribution of correlation $R^2$ among multiple genes were compared to $R^2$ values derived from null distribution using Fisher's transformation through a modified R script originally from the Im lab (https://gist.github.com/hakyim/a925fea01b365a8c605e).

Narrow-sense heritability ($h^2$) of phenotype from imputed GWAS data from unrelated patients was calculated using the GREML-LDMS method [19] from the Genome-wide Complex Trait Analysis (GCTA) software [20], v1.93.0beta.

For hierarchical clustering, signed -log10p-value with sign of association beta coefficient as indicator of expression change direction were compiled for genes significantly associated to disease phenotype from multiple tissue data sets. Clustering heatmaps were generated using the Bioconductor R package, *ComplexHeatmap* [21] (additional details provided in the S1 Methods in S4 File). Manhattan plots of GWAS data and imputed gene expression phenotype associations were generated using the R package, *qqman* [22], and *ggplot2* [23]. GWAS p-values of relevant SNPs were formatted as bedGraph files, and visualized on the UCSC genome browser (http://genome.ucsc.edu/) as custom annotation tracks against appropriate reference genomes.

Pre-ranked Gene Set Enrichment Analysis [24] against several collection of gene sets and pathways were performed with both PrediXcan and TWAS platforms using the Bioconductor R package *fgsea* [25]. The ranks were based on the -log10 of the maximal p-value between the 2 meta-analysis methods applied for each platform. In addition, candidate genes were functionally categorized using Gene Ontology (GO) terms [26], and Reactome annotations [27], coupled with expert review of the literature.

## Results

### Predictive models for genetic regulation of gene expression using training data from CF cohorts

To build predictive models of genetic regulation of gene expression with training data from CF patients, we adapted the PredictDB_Pipeline for GTEx_v7 to work with CF genotype and gene expression data from both LCL [11] and nasal epithelial biopsy [12] data sets. The performance of the predictive models was evaluated by the correlations between predicted and observed gene expression, and genes were filtered at minimal performance suggested by PredictDB. The number of imputable genes (as defined by prediction $R^2 > 0.01$ and p-value < 0.05), including protein-coding, lincRNA, and pseudogenes, from nasal epithelial biopsy data set consisting of 132 training samples was 2,881; while that from 753 LCL data set was 5,299. As shown in S1 Fig in S4 File, the predicted vs observed $R^2$ from both data sets are significantly higher than expected from null distribution, with the average $R^2$ of 0.11 and 0.072 for imputable genes from nasal epithelial biopsy and LCL models, respectively, comparable to reported models based on GTEx data sets [9]. These $R^2$ values suggest the existence of a substantial number of genes whose expression can be partially explained by genetic variants. The degree of $R^2$ deviation from null between nasal epithelial biopsy (n = 132) and LCL (n = 753)

models reflect the sample size difference between them, since sample size and quality of training data are critical factors that determine the performance of the predictive models and the number of predictable genes [10]. Our nasal epithelial biopsy models are comparable to GTEx RNA-seq data sets from PrediXcan, while our LCL microarray data set yielded fewer than expected number of imputable genes (S2 Fig in S4 File).

We investigated correlations of our CF LCL model predictions with those of GTEx on the same set of patients. The numbers of imputed genes that passed respective prediction filters are 5,299 from CF LCL, and 3,039 from GTEx Cells_EBV-transformed_lymphocytes (i.e. LCLs), with overlap of 1,623 genes by ENSEMBL gene_id. The correlation of the 1,623 genes between the 2 data sets were calculated and compared to expected $R^2$ distribution from null (S3 Fig in S4 File). The mean $R^2$ value among 1,623 genes is 0.51, i.e. the two imputed gene expression data sets are highly correlated, suggesting similar genetic regulation of gene expression in the same cell type in independent training data sets. Also as reported, there is significant cross predictability of the models between different tissues [9], and the correlation between imputed gene expression from CF LCLs, and GTEx lung tissue, among 2,552 genes predicted in both data sets, are also significantly above null, with mean $R^2$ of 0.40 (S3 Fig in S4 File).

## Association of genetically regulated gene expression to CF lung disease severity

Association testing of imputed gene expression from a total of 51 tissues (2 CF, 48 GTEx, and DGN whole blood) were performed using robust regression against the quantitative lung function score, KNoRMA, and results from all tissues were used in meta-analysis as described in methods (Fig 1). The meta-analyses resulted in 245 candidate modifier genes from PrediXcan by consistent p-value < 0.01 from 2 meta-analyses (HMP.PrediXcan, EBM.PrediXcan) and 186 candidate genes utilizing GWAS summary statistics and TWAS/FUSION meta-analyses (HMP.TWAS, OMNIBUS.TWAS), giving a combined candidate list of 379 unique genes (S1 File). Using a threshold of p-value < 0.01 across all 4 meta-analyses, 52 consensus CF lung disease modifier genes were defined (Figs 2 and 3, Table 1). Several key features of these 52 consensus genes are highlighted in Fig 2. First, there is a general agreement between PrediXcan (left panel) and TWAS (right panel) in terms of direction (color) and strength (intensity) of the association of imputed gene expression to lung disease severity. Second, more than half (28 out of 52) of the consensus genes were located within 1 Mb of the 5 autosomal GWAS signals. Third, the direction of the predicted effect of gene expression as it relates to the lung disease phenotype varies across genes (blue versus red) and is relatively consistent across tissues, with rare exceptions (discussed below). Fourth, association signal is often centered around GWAS loci and with genes imputed across many tissues, although there are exceptions. Many of these genes have relevance to known features of CF pathogenesis (see citations in Table 1), and the direction of imputed gene expression change reflects the direction of alleles and prediction weights of SNPs in the predictive models. Among the 52 consensus modifier genes, the correlation coefficient between average effect sizes from multiple tissues between PrediXcan and TWAS is r = 0.83 ($R^2$ = 0.69, S4B Fig in S4 File), while that from the maximal multi-tissue p-values of PrediXcan and TWAS, is r = 0.68 ($R^2$ = 0.46, S4C Fig in S4 File). As shown by the color of the heatmaps in Fig 2, most of the consensus modifier genes are similar in change of direction relative to KNoRMA across multiple tissues with strongest signals from chr5 and chr6 GWAS loci, such as *EXOC3*, and *HLA-DRB1*, respectively. However, there are some exceptions, such as *TPPP* and *MET*, where genetic regulations of expressions associate to KNoRMA with different direction in different tissues. For example, *TPPP* is predicted to be

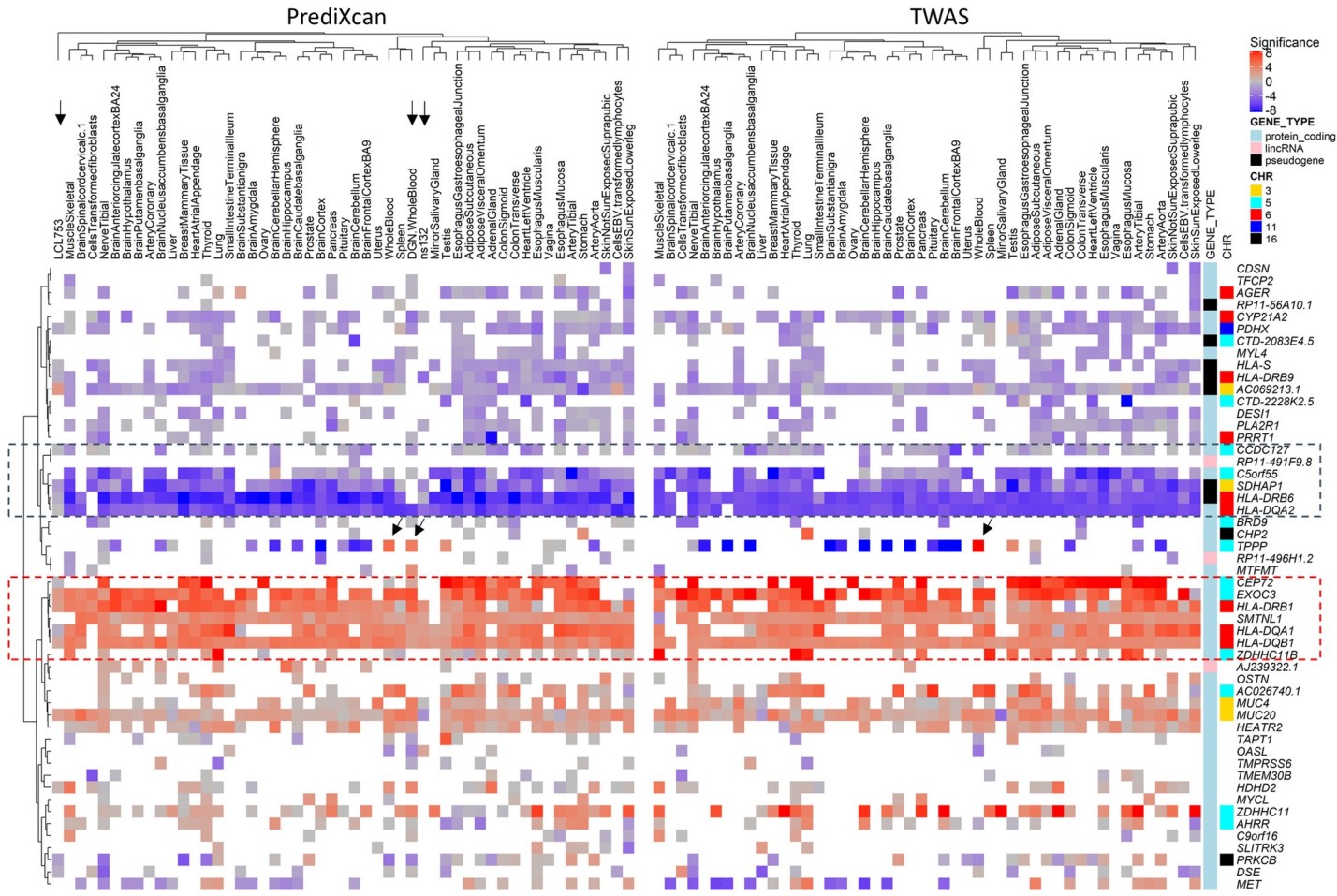

**Fig 2. Hierarchical clustering of genes whose imputed expression are associated with CF lung disease severity.** Consensus modifier genes (n = 52) were determined as p-value < 0.01 from all 4 meta-analyses of multiple tissue association testing described in methods, and the -log10(p-values) were clustered and represented as a heatmap with red-grey-blue color scale. The color represents direction of predicted expression change, with red indicates "protective", or increased expression with increasing KNoRMA (milder lung disease), and blue, "harmful", or increased expression with decreasing KNoRMA (more severe lung disease), and the intensity reflects the significance (p-values) of the association. White cells in heatmap indicate missing data, where the genes were not well predicted from the relevant tissues. The vertical color columns on the right indicate type of gene and chromosome near GWAS loci. The genes were clustered based on results from PrediXcan (left heatmap), and the order of the genes were kept the same for TWAS (right heatmap). Key patterns of negative and positive associations to KNoRMA across multiple tissues in the heatmap are highlighted by the dashed boxes. Arrows on top of the left heatmap identify the additional tissues over the 48 GTEx tissues common to both platforms, and arrows in the middle of the heatmaps show the results from whole blood tissues for *TPPP*.

increased in milder patients (higher KNoRMA values) from both GTEx and DGN whole blood, while the opposite is predicted from other tissues.

As expected from published PrediXcan and TWAS applications to other diseases [76, 77], many genes associated with CF lung disease severity are around the reported genome-wide significant loci from GWAS (red squares in Fig 3, and Table 1A), but there are also significant genes elsewhere (blue triangles in Fig 3, and Table 1B), including *MET* ~700 kb upstream of *CFTR* on chr7, *TAPT1* on chr4, and *HEATR2* on chr7 to name a few. This provides evidence for significant association with SNPs outside the GWAS significant loci and/or combinatorial signals from the multiple SNPs used in predictive models. Further, the genome-wide significant signal by fixed-effect meta-analysis p-value on chr16 (Fig 3C, S5 Fig in S4 File), which was not reported in the GWAS publication due to multiple hypothesis testing penalty [1], was brought to attention by gene expression imputation for *CHP2* and *PRKCB* (Fig 3A and 3B).

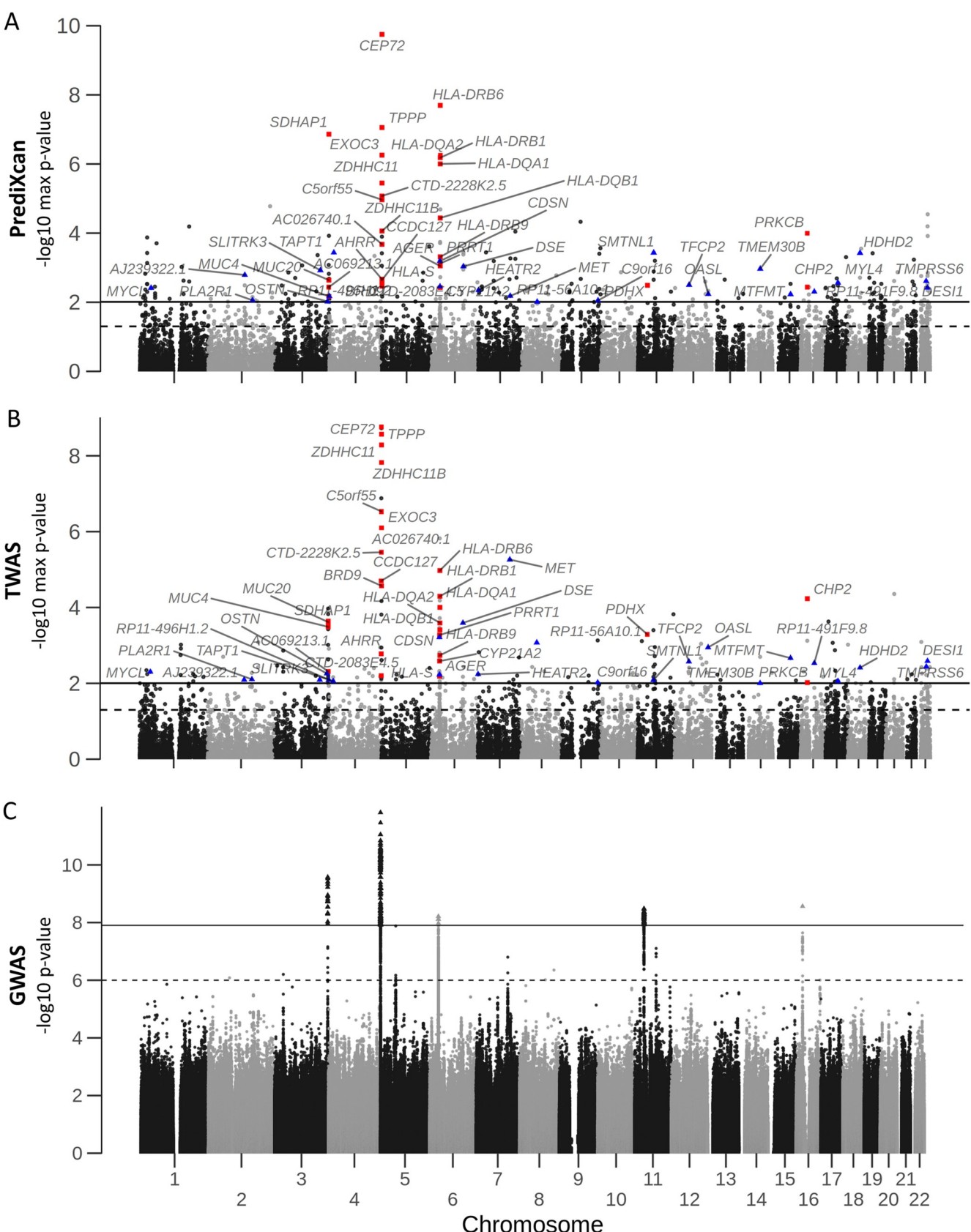

**Fig 3. Manhattan plots of CF lung disease association p-values from gene expression imputation and GWAS.** Maximal p-values between 2 meta-analyses from imputed gene expression to KNoRMA by PrediXcan and TWAS were used in the Manhattan plots A and B respectively. The 28 consensus modifier genes within 1 Mb of 5 autosomal GWAS signals (red squares), and those not near GWAS signals (blue triangles) are labeled. Panel C represents GWAS p-values from the updated imputation [78] by fixed-effect meta-analysis performed according to the GWAS study [1]. The solid lines correspond to genome-wide significant p-value of 0.01 (for imputed expression, A and B) or $1.25 \times 10^{-08}$ (for GWAS, C), while the dashed lines represent the suggestive p-value of 0.05 (for imputed expression) or $1 \times 10^{-06}$ (for GWAS).

To globally compare GWAS association with imputed expression association, available SNP GWAS association p-values for the *cis*-SNPs used as predictive variables, were retrieved for all imputable genes of PrediXcan predictive models of all 48 GTEx tissues. Minimal SNP p-values in predictive models of a gene were compared to the maximal association p-value between HMP.PrediXcan and EBM.PrediXcan for the same gene to CF lung disease severity from imputed expression (Fig 4). The correlation coefficient of the minimal GWAS -log10 p-values with PrediXcan maximal association p-values over the > 25,000 imputable genes is highly significant, with r = 0.19 ($R^2$ = 0.036, Fig 4). Similarly, mean SNP GWAS p-value and imputed expression p-value among these genes are also significantly correlated with r = 0.13 ($R^2$ = 0.017, S6 Fig in S4 File). As indicated above, examples of significant associations from imputed gene expression from regions where no genome-wide significant SNPs were identified from the GWAS include *DESI1*, *HEATR2*, *OASL*, *SLITRK3*, *TAPT1*, *etc*. (Fig 3, and Table 1B).

The integration of SNP association to lung disease phenotype (GWAS) and imputed eQTL signals can be illustrated by examining the SNPs utilized in the models to predict expression for the chr11 locus, as shown in Fig 5 (and S7 Fig in S4 File). Combining predictive variables (SNPs) from multiple GTEx tissue models, and among SNPs with significant GWAS p-values of $< \times 10^{-07}$ [top annotation track in Fig 5 (zoom-in view), S7 Fig in S4 File (full region)], only 1 SNP (among 50 in all *EHF* models) was used to impute *EHF* expression, and only 2 SNPs (among 759 in all *APIP* models) were used for *APIP*. In contrast, 20 of the significant SNPs were predictive for *PDHX*, which in turn translated into significant lung disease associations of imputed gene expression for *PDHX* (Figs 2 and 3, and Table 1), but not *EHF* and *APIP*, even though *EHF* and *APIP* are closest to the GWAS signal. Similarly, imputed eQTL data help to point to genes regulated by SNPs at other regions (S8-S12 Figs in S4 File) and suggest the direction of genetically regulated expression change in regard to phenotype trait (Table 1).

### Gene set enrichment analyses and functional categories of candidate CF lung disease modifier genes

Gene set (pathway) enrichment analyses (GSEA) were performed based on protein-coding genes pre-ranked by the maximal p-value between the 2 multi-tissue meta-analyses for each analysis platform, PrediXcan and TWAS. Since all imputed protein-coding genes of PrediXcan (n = 16,431) and TWAS (n = 13,685) were ranked, GSEA can uncover concerted association of gene set or pathway members with CF lung disease (S1, S2 Tables in S4 File). Apart from the usual suspects of immune and vesicle trafficking processes and pathways reported in previous publications, including a large number of pathways dominated by HLA genes [11, 12, 79, 80], some highly specific, pathogenically relevant processes were also enriched, with examples of "Interferon-gamma-mediated signaling pathway" from GO biological process, "Defective CFTR causes cystic fibrosis" and "Antimicrobial peptides" from Reactome pathway, and "Asthma" from KEGG pathway shown in Fig 6 (and in S1, S2 Tables in S4 File).

Alternatively, we looked for overlaps between the 379 potential candidate modifiers of CF lung disease (described above) and CF relevant-biological categories, many of which are represented by GSEA analyses. Using GO and Reactome annotations, coupled to key functional

**Table 1. Consensus 52 CF lung disease modifier genes.**

| Gene | Gene type | chr | p-value (max) | Direction* | | CF-related citations |
|------|-----------|-----|---------------|-----------|---|----------------------|
| **A: Genes in regions of GWAS association ordered by chromosome** | | | | | | |
| MUC20 | protein coding | 3 | $8.1 \times 10^{-03}$ | Protective (0.014;2.44) | Mucus barrier | |
| MUC4 | protein coding | 3 | $5.9 \times 10^{-03}$ | Protective (0.011;2.1) | Epithelial membrane mucin; possible regulation by CFTR | [28] |
| SDHAP1 | pseudogene | 3 | $2.3 \times 10^{-04}$ | Harmful (-0.021;-4.1) | | |
| AC069213.1 | pseudogene | 3 | $4.9 \times 10^{-03}$ | Harmful (-0.012;-2.06) | | |
| AC026740.1 | protein coding | 5 | $3.1 \times 10^{-04}$ | Protective (0.01;2.97) | | |
| AHRR | protein coding | 5 | $3.7 \times 10^{-03}$ | Protective (0.003;0.97) | Aryl hydrocarbon receptor | [29, 30] |
| BRD9 | protein coding | 5 | $1.3 \times 10^{-04}$ | Harmful (-0.002;-3.95) | Lysine-acetylated histone binding, chromatin organization; important in small lung cell cancers | |
| C5orf55 | protein coding | 5 | $4.7 \times 10^{-05}$ | Harmful (-0.02;-3.95) | EXOC3 antisense | |
| CCDC127 | protein coding | 5 | $5.8 \times 10^{-03}$ | Harmful (-0.006;-1.83) | Regulates HSP70 gene expression; HSP70 is involved in CFTR processing | [31, 32] |
| CEP72 | protein coding | 5 | $1.8 \times 10^{-09}$ | Protective (0.019;5.66) | Microtubule-organizing, organelle, centrosome; required for cilia formation; microtubules and cilia important for CF pathophysiology | [33–39] |
| CTD-2083E4.5 | pseudogene | 5 | $6.3 \times 10^{-03}$ | Harmful (-0.007;-1.8) | | |
| CTD-2228K2.5 | protein coding | 5 | $1.6 \times 10^{-05}$ | Harmful (-0.01;-2.99) | | |
| EXOC3 | protein coding | 5 | $3.5 \times 10^{-06}$ | Protective (0.028;4.86) | Exocytosis, epithelial polarity; interaction with actin cytoskeletal remodeling and vesicle transport machinery; components of exocyst complex required for intracellular bacteria clearance from cells; regulates MUC5AC secretion induced by neutrophil elastase in human airway epithelial cells | [40] |
| TPPP | protein coding | 5 | $1.0 \times 10^{-07}$ | Harmful (-0.012;-4.08) | Microtubule bundle; microtubules associated with CFTR-related pathogenic processes (see CEP72 above) | [41–47] |
| ZDHHC11 | protein coding | 5 | $9.4 \times 10^{-06}$ | Protective (0.005;4.41) | Palmitoylation, ER, Golgi protein targeting; mediator of DNA virus response | [48] |
| ZDHHC11B | protein coding | 5 | $1.1 \times 10^{-04}$ | Protective (0.003;4.13) | Palmitoylation, ER, Golgi protein targeting | |
| AGER | protein coding | 6 | $6.5 \times 10^{-03}$ | Harmful (-0.007;-2.39) | Associated with pathogen load, inflammation, and hypoxia in CF | [49–51] |
| CYP21A2 | protein coding | 6 | $2.6 \times 10^{-03}$ | Harmful (-0.01;-2.39) | Steroid hydroxylase, congenital adrenal hyperplasia; Cytochrome P450 superfamily; required for the synthesis of steroid hormones including cortisol and aldosterone. | |
| HLA-DQA1 | protein coding | 6 | $1.0 \times 10^{-04}$ | Protective (0.026;3.84) | Ancestral allele 8.1, CF delayed onset infection; potential CF modifier in pancreas and liver | [52, 53] |
| HLA-DQA2 | protein coding | 6 | $2.5 \times 10^{-04}$ | Harmful (-0.049;-4.76) | Ancestral allele 8.1, CF delayed onset infection; highly conserved in contrast to some other HLA genes | [54, 55] |
| HLA-DQB1 | protein coding | 6 | $3.9 \times 10^{-04}$ | Protective (0.04;3.48) | Ancestral allele 8.1, CF delayed onset infection; potential CF modifier in pancreas and liver | [52, 53, 56] |
| HLA-DRB1 | protein coding | 6 | $5.1 \times 10^{-05}$ | Protective (0.024;3.61) | Ancestral allele 8.1, CF delayed onset infection; associated with allergic and T(H)-1 like responses | [52, 56–58] |
| HLA-DRB6 | pseudogene | 6 | $1.1 \times 10^{-05}$ | Harmful (-0.052;-4.67) | Ancestral allele 8.1, CF delayed onset infection | |
| HLA-DRB9 | pseudogene | 6 | $1.8 \times 10^{-03}$ | Harmful (-0.017;-2.77) | Ancestral allele 8.1, CF delayed onset infection | |

(*Continued*)

**Table 1.** (*Continued*)

| Gene | Gene type | chr | p-value (max) | Direction* | | CF-related citations |
|------|-----------|-----|---------------|------------|--|----------------------|
| *PRRT1* | protein coding | 6 | 5.3x10^{-04} | Harmful (-0.01;-2.39) | Post synaptic membrane | |
| *PDHX* | protein coding | 11 | 3.1x10^{-03} | Harmful (-0.011;-2.01) | Mitochondrial glycolysis, congenital lactic acidosis; pyruvate dehydrogenase, an enzyme complex linking glycolysis with downstream oxidative metabolism, represents a key location where regulation of metabolism occurs; PDHX is a key structural component of this complex and is essential for its function; involved in glucose metabolism so associated with oxidative responses | |
| *CHP2* | protein coding | 16 | 1.9x10^{-03} | Protective (-0.002;0.74) | Cellular pH regulation, plasma membrane Na+/H+ exchangers required as an obligatory binding partner for ion transport | |
| *PRKCB* | protein coding | 16 | 9.6x10^{-03} | Harmful (-0.002;-0.1) | Adaptive immunity, B cell activation; Linked to CFTR mRNA expression, Regulation of autophagy via sensing of mitochondrial energy status | [59, 60] |
| **B: Genes in regions of no prior association (in this cohort of subjects) ordered by chromosome** | | | | | | |
| *MYCL* | Protein coding | 1 | 5.0x10^{-03} | Protective (0.006;2.28) | Dis-regulation associated with lung and other cancers | [61] |
| *AJ239322.1* | lincRNA | 2 | 8.1x10^{-03} | Protective (0.007;2.74) | | |
| *PLA2R1* | Protein coding | 2 | 8.8x10^{-03} | Harmful (-0.008;-2.11) | Potential target in asthma | [62, 63] |
| *RP11-496H1.2* | lincRNA | 3 | 8.0x10^{-03} | Harmful (-0.004;-2.43) | | |
| *OSTN* | Protein coding | 3 | 9.5x10^{-03} | Protective (0.005;1.82) | | |
| *SLITRK3* | protein coding | 3 | 8.1x10^{-03} | Protective (0.002;2.32) | Synaptic membrane adhesion; involved in GABAergic synapse formation; recent evidence of GABAergic control of mucous cell differentiation in human airway epithelium | [64, 65] |
| *TAPT1* | protein coding | 4 | 8.7x10^{-03} | Harmful (-0.0004;-0.36) | Cilia basal body, centrosome; associated with lung function decline in smokers | |
| *DSE* | Protein coding | 6 | 9.2x10^{-04} | Harmful (-0.006;-1.51) | Dermatan sulfate is part of proteoglycans that are involved in many biological processes, such as cancer, immunity, and defect can cause Ehlers-Danlos syndrome, which may lead to hypoplasia of the lung | [66, 67] |
| *CDSN* | protein coding | 6 | 6.1x10^{-04} | Harmful (-0.015;-3.75) | Cell adhesion, skin morphogenesis; epithelial cell differentiation | |
| *HLA-S* | pseudogene | 6 | 5.9x10^{-03} | Harmful (-0.019;-2.5) | | |
| *HEATR2* | protein coding | 7 | 5.8x10^{-03} | Protective (0.011;2.21) | *DNAAF5* (alias), motile cilia, necessary for assembly of the ciliary motile apparatus | [68, 69] |
| *MET* | protein coding | 7 | 7.2x10^{-03} | Harmful (-0.006;-0.92) | Genetic marker, *CFTR* mutation | [70] |
| *RP11-56A10.1* | pseudogene | 8 | 7.4x10^{-03} | Harmful (-0.007;-3.16) | | |
| *C9orf16* | protein coding | 9 | 9.6x10^{-03} | Protective (-0.0001;0.34) | | |
| *SMTNL1* | protein coding | 11 | 8.2x10^{-03} | Protective (0.022;3.08) | Muscle contraction | |
| *OASL* | protein coding | 12 | 4.6x10^{-03} | Harmful (-0.004;-1.8) | Antiviral, inhibits RSV | [71–73] |
| *TFCP2* | protein coding | 12 | 2.7x10^{-03} | Harmful (-0.003;-2.56) | Transcription factor, alpha-globin, inflammatory response | |
| *TMEM30B* | protein coding | 14 | 9.9x10^{-03} | Harmful (-0.002;-0.61) | Phospholipid translocation | |
| *MTFMT* | protein coding | 15 | 5.6x10^{-03} | Harmful (-0.003;-1.61) | Mitochondrial translation, required for mitochondrial function/oxidative phosphorylation | |

(*Continued*)

**Table 1.** (Continued)

| Gene | Gene type | chr | p-value (max) | Direction[*] | | CF-related citations |
|------|-----------|-----|---------------|-----------|---|----------------------|
| *RP11-491F9.8* | lincRNA | 16 | 7.5x10$^{-03}$ | Harmful (-0.015;-3.25) | | |
| *MYL4* | protein coding | 17 | 8.7x10$^{-03}$ | Harmful (-0.005;-2.69) | Actin filament binding, atrial fibrillation | |
| *HDHD2* | protein coding | 18 | 3.9x10$^{-03}$ | Protective (0.003;1.51) | | |
| *DESI1* | protein coding | 22 | 2.6x10$^{-03}$ | Harmful (-0.009;-2.84) | Proteolysis; desumoylating isopeptidase; SUMO paralogues determine fate of wild-type and mutant CFTR protein | [74] |
| *TMPRSS6* | Protein coding | 22 | 3.6x10$^{-03}$ | Harmful (-0.0004;-0.36) | AKA matriptase-2, variants associated with iron refractory iron deficiency anemia | [75] |

[*]**Direction** defined as: **Harmful** (PrediXcan beta coefficient; TWAS zscore): Increased expression correlated with worse lung disease (decreased KNoRMA), or

**Protective** (PrediXcan beta coefficient; TWAS zscore): Increased expression correlated with milder lung disease (better KNoRMA)

categories identified with CF relevance (Table 1), we classified 149 of the 379 candidate genes into 11 functional categories (Table 2).

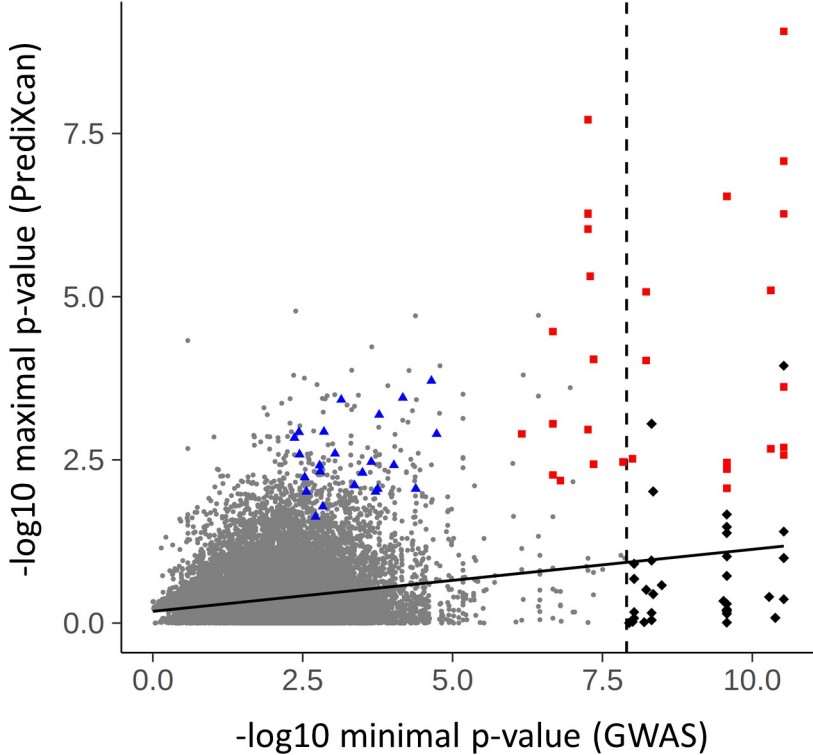

**Fig 4. Correlation of imputed gene expression association from PrediXcan and minimal GWAS association p-values.** Maximal p-values between HMP and EBM meta-analyses of CF lung disease associations from imputed gene expression (PrediXcan) for 26,750 genes from 48 GTEx tissues are plotted against minimal GWAS SNP p-values per gene among all *cis*-SNPs used in predictive models. The 52 consensus modifier genes are highlighted in red squares (near GWAS loci) and blue triangles (novel), while genes with minimal GWAS SNP p-values < x10$^{-08}$ (dashed vertical line), but not among the 52, are highlighted in black diamonds. Solid line represents linear regression.

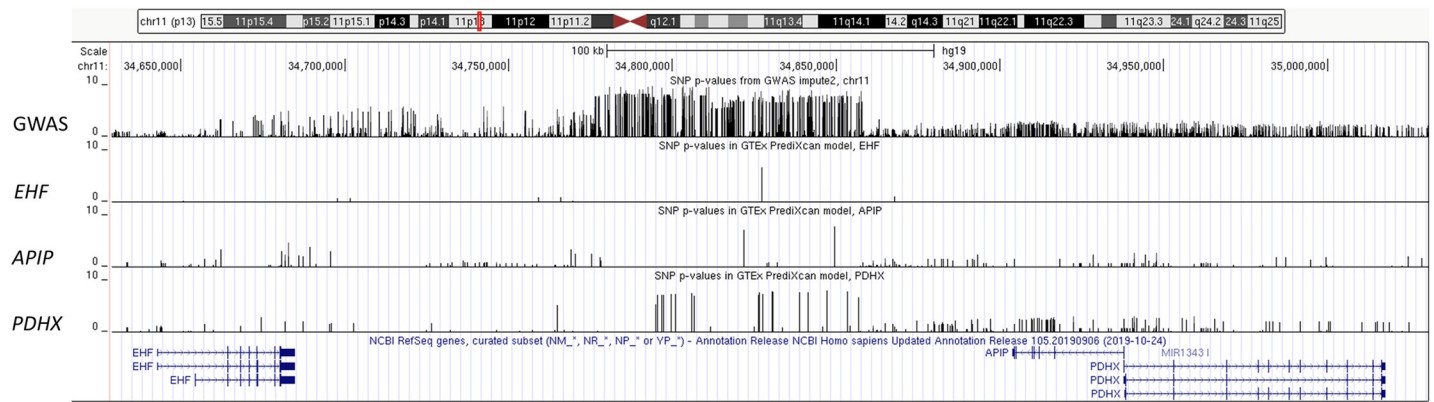

**Fig 5. Comparison of predictive model SNPs at chromosome 11 GWAS locus.** The -log10 p-values from GWAS analysis were retrieved for *cis*-SNPs in viable PrediXcan predictive models from 48 GTEx tissues for *EHF*, *APIP*, and *PDHX*. These p-values were formatted as bedGraph files and displayed through the UCSC genome browser (http://genome.ucsc.edu/) as custom annotation tracks, with vertical scales set between 0 and 10. The screenshot of the genome browser shows from top to bottom: GWAS SNP p-values, SNPs used in *EHF* gene expression imputation model, those for *APIP*, *PDHX*, and gene annotation from NCBI RefSeq genes.

## Allele bias of gene expression estimation may confound interpretation of hyper-variable genes, such as HLAs

Many HLA genes appear to be strongly regulated genetically, as reflected by variance explained or $R^2$ of the predictive models (S3, S4 Tables in S4 File) and HLA-dominated pathways are highly significant in our previous gene expression association studies [11, 12]. However, since gene expression quantification relies on mapping of RNA-seq reads to genome/transcriptome sequences, expression levels may be biased towards the reference allele, especially for the hypermorphic HLA genes [81, 82]. To assess influences of allele bias on gene expression quantification and trait association, we compared different strategies of RNA-seq read mapping from our nasal epithelial biopsy RNA-seq data set. In addition to the standard protocol of mapping to the primary reference genome assembly, we also adopted an alternative mapping strategy to include additional alternative genome assemblies as suggested [82], and incorporated common variance information (http://ccb.jhu.edu/hisat-genotype) from dbSNP v150 (S1 Methods in S4 File). As shown in S13 Fig in S4 File, the correlation and spread of expression estimates are similar for selected HLA Class II genes, between AltHapAlignR [82] and default gene counts (S13A-S13D Fig in S4 File), and alternative mapping FPKM (Fragments Per Kilobase per Million) and standard mapping FPKM (S13E-S13H Fig in S4 File). When the bias-corrected alternative gene expression quantification was used in predictive model building, gene expression imputation, and trait association testing, the results were dramatically different for some genes, such as *HLA-DQA1* and *HLA-DRB1*, where the direction of predicted expression changes in regard to lung function are opposite between different mapping strategies (Fig 7A). The number of genes that can be predicted by *cis*-SNPs among the bias-corrected training set, compared to the standard protocol that predicted 2,881 genes (S2 File), increased by >1,000 to 4,263 (S3 File), with only 1,379 overlap between them. These findings suggest that allele bias associated with commonly employed gene expression estimation pipelines can confound phenotype association testing, resulting in misinterpretation of genetic modulation of phenotype apparently via gene expression regulation.

## Discussion

We have applied gene expression imputation to mine the CF gene modifier GWAS data set and extracted 379 potential and 52 consensus CF lung disease modifier candidates. The

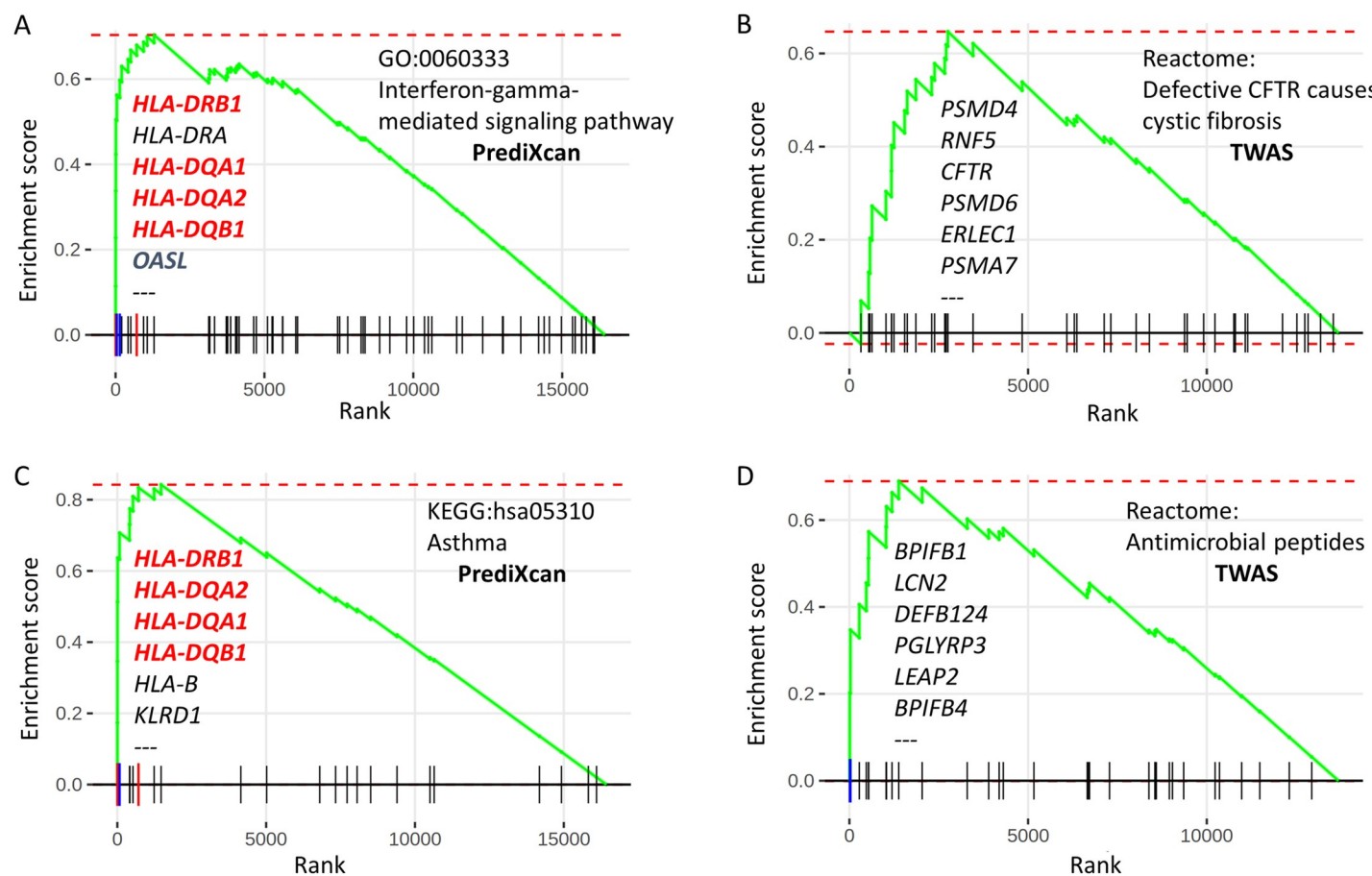

**Fig 6. Gene set enrichment plots.** Gene set enrichment analyses (GSEA) were performed and enrichment plots were generated for selected gene sets using the Bioconductor R package, *fgsea*. For each enrichment plot, the horizontal black line at the bottom represent p-value ranks of protein-coding genes with most significant p-value rank on the left. The vertical bars represent individual genes in a gene set and their ranks. The green curves represent the cumulative enrichment score (ES), and the red horizontal dashed lines denote minimal (often 0) and maximal scores. Listed genes represent the leading edge with increasing ES, that contribute to the overall enrichment of the gene set. Panel A and C are GSEA results from PrediXcan platform, while B and D from TWAS. Particular gene sets shown are from GO biological process (A), and Biosystems (C–KEGG, B, D–Reactome).

imputation techniques leveraged GTEx integrative training data sets from 48 human tissues [5], a large RNA-seq data set from whole-blood (DGN) [13], and our own CF gene expression data sets from nasal epithelial biopsy [12] and LCL [11] samples. Twenty eight of the 52 consensus genes are within 1 Mb of the 5 autosomal genome-wide significant loci [1], while 24 consensus modifier genes were not identified in GWAS. Overall, integration of GWAS with eQTL data through gene expression imputation highlighted some candidate modifier genes (Figs 3 and 4, red squares), and diminished potential roles of others (Fig 4, black diamonds) around GWAS loci, as well as uncovered modifiers outside GWAS loci (Figs 3 and 4, blue triangles). Disease phenotype association testing of the imputed gene expression also predicted the direction of genetically regulated gene expression changes relative to CF lung disease severity, which provides guidance on mechanism of disease modification, and potential intervention strategies. By using independently developed divergent approaches, we sought to balance sensitivity by combining the results from multiple tissues and platforms, and robustness by consensus of the findings between PrediXcan and TWAS. The consensus and potential CF lung disease modifier genes were then evaluated by biological context through literature review and gene set enrichment analyses.

**Table 2. Functional categories of significant genes (n = 149 out of 379) relevant to CF pathophysiology**[*]**.**

| Category | Genes |
|---|---|
| Immunity/ infection/inflammation | *AGER, AHRR, EXOC3, HLA-DQA1, HLA-DQA2, HLA-DQB1, HLA-DRB1, MET, MUC20, MUC4*, OASL, *PRKCB*, TFCP2; ADAM, AMBP, AP1S1, ATP6V0D2, AZU1, BPIFA1, BPIFB1, BTNL2, C2, CEACAM6, CFH, DDX60, EFNB3, FGF20, FRK, GAN, HLA-B, HLA-DQB2, HLA-DRA, IGSF5, JMJD6, LCN2, METTL7A, MEX3C, MME, NDC1, NFAM1, NPY5R, ORMDL3, PIK3R2, PRG2, RAC2, RORC, SLC3A2, SLFN13, SMAD4, SPG21, TFRC, TREX1, UBE2Z, VAV3, YTHDF2, ZFP36L2, ZYX* |
| Mucociliary clearance | *C5orf55, CEP72, EXOC3*, HEATR2, *MUC20, MUC4*, SLITRK3, TAPT1, *TPPP*; AK8, ARL3, CEP120, ICK, IFT74, MYO3B, NUBP1, PROM1 |
| Glycosylation | *AGER, MUC20, MUC4*; A4GALT, ARFGAP3, GOSR1, NOTCH4, PIGO, PIGW, SERP1, ST3GAL6, TRAPPC2L, XXYLT1 |
| Viral/virus | *HLA-DQA1, HLA-DQA2, HLA-DQB1, HLA-DRB1*, OASL; AMBP, ATP6V0D2, AZU1, BPIFA1, CFH, DDX39B, DDX60, EFNB3, HLA-B, HLA-DRA, LCN2, NDC1, PIK3R2, RAC2, RPS10, SLFN13, STMN1, TFRC, TREX1, ZYX* |
| Mitochondria | *MTFMT, PDHX*; BIK, DDAH2, HIGD2A, HRK, MMAA, MTFR1L, MTG1, MYO19, NDUFAF6, NRF1, RAC2, SDHA, TARS2, TDRKH, TIMM10* |
| ER/Golgi | *DSE, EXOC3*, TAPT1, TMEM30B, *ZDHHC11, ZDHHC11B*; A4GALT, AKR7A2, AP1S1, ARFGAP3, ARL3, BSCL2, CPD, CUX2, GOSR1, IER3IP1, METTL7A, NOTCH4, ORMDL3, PIK3R2, SERP1, STC2, TFRC, TRAPPC2L, XXYLT1* |
| Ubiquitination | GAN, GNA12, MEX3C, PIAS2, SMAD4, TNK2, UBE2Q2P1, UBE2Z, UFD1L |
| Lipid | *AHRR, CYP21A2, PLA2R1*, TMEM30B, *ZDHHC11, ZDHHC11B*; *A4GALT*, APOC2, BSCL2, CYP21A2, FADS3, GLTP, GNA12, JAZF1, LDLRAP1, MED19, MMAA, NCOA3, NRF1, NRIP1, ORMDL3, OSBPL10, PIGO, PIGW, PIK3R2, PLA2R1, PNLIPRP3, SERINC1, SOAT1, THRB, TREX1* |
| CFTR interactome | RAC2, SDHA, TARS2, YTHDF2 |
| Transcription factors | AATF, FOXP2, NCOA3, NEAT1, NRF1, NRIP1, PIAS2, RORC, SMAD4, TFCP2, THRB |
| Cytoskeleton/ microtubule | *CEP72*, MET, SMTNL1, TAPT1, *TPPP*; ADD3, ARL3, AUNIP, CEP120, GAN, GAS2L3, GNA12, ICK, IFT74, MAST3, MYO19, NUBP1, PACSIN2, PDLIM3, PIK3R2, POC5, RAC2, SMTNL1, SPATC1L, STMN1, TAPT1, TPPP, VILL, ZYX* |

[*]Alphabetical listing for 28 (of 54) consensus genes near (bold) and outside (underlined) GWAS loci (between TWAS and PrediXcan, Table 1); remaining genes (n = 121, alphabetically listed) are from the other 327 significant candidate modifier genes (S1 File)

The usefulness of defining the relationship of SNP association to the imputed gene expression association to phenotype, deduced through independent eQTL data sets, can be illustrated at the chr11 locus (Fig 5, S7 Fig in S4 File). Although *EHF* and *APIP* are the nearest genes to the intergenic chr11 GWAS locus with significant lung disease association p-values, *PDHX* is best predicted to be regulated by SNPs in the region based on current gene expression data. These results do not rule out developmental and other cell/tissue-specific mechanisms not assessed, by which *EHF* and *APIP* may modify CF lung disease process. Nevertheless, *PDHX* is a critical gene in mitochondrial energy metabolism (OMIM: 245349) that should be investigated further, since many additional candidate modifiers related to mitochondrial function were also identified in this study (Table 2).

Examples at other genomic loci are also informative (S8-S12 Figs in S4 File). The strongest GWAS signals on chr5 supported by gene expression imputation (Fig 3) contain 3 genes, *CEP72, TPPP*, and *EXOC3* (Figs 2 and 3, S9 Fig in S4 File, Table 1) involved in microtubule organization and exocytosis. *MUC4* and *MUC20* are significant at chr3 (S8 Fig in S4 File), and

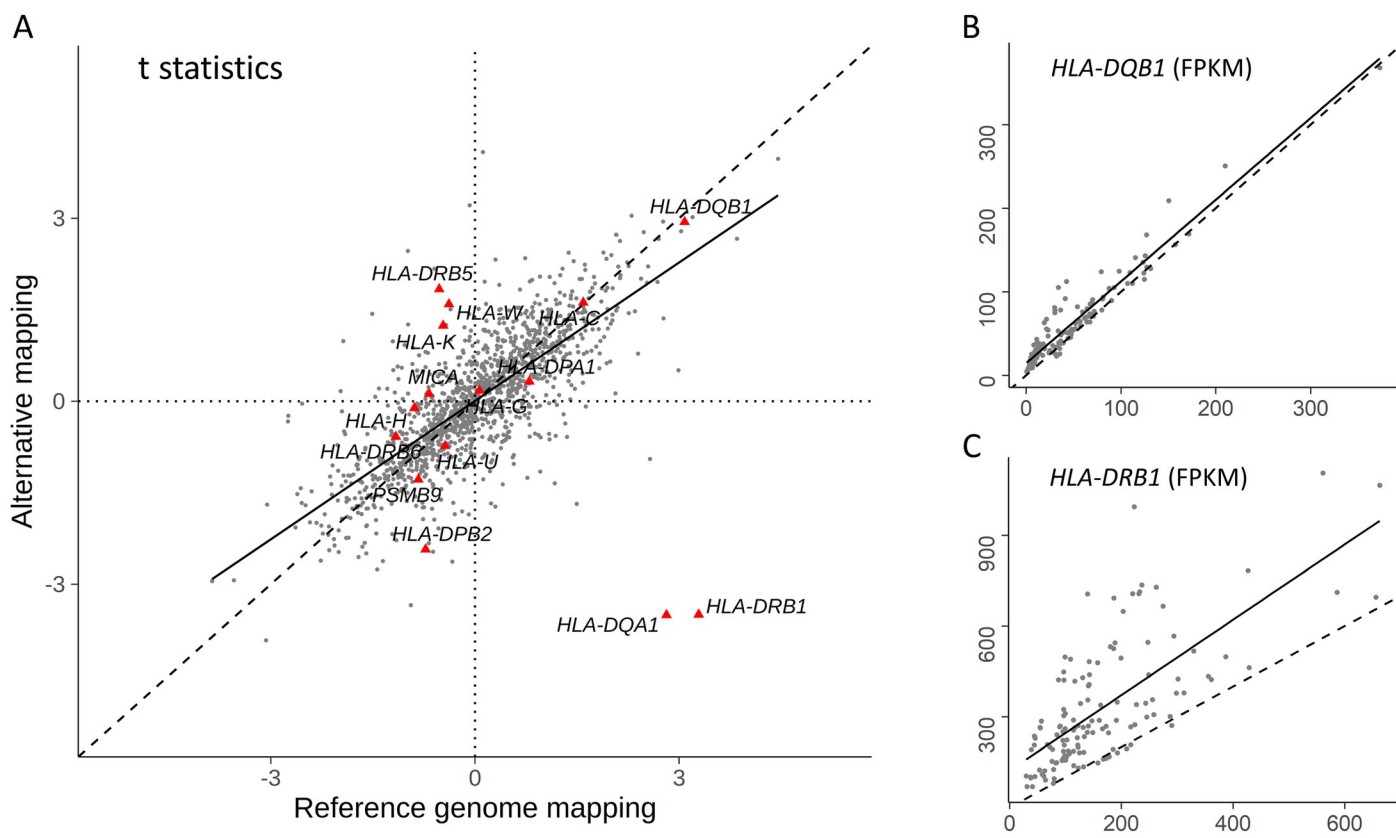

**Fig 7. Effect of allele bias on gene expression quantification and disease phenotype association in CF nasal epithelial biopsy RNA-seq data set.** Comparison of CF lung disease (KNoRMA) association t statistics between different mapping protocols among 1,379 common imputable genes by respective predictive models among 5,634 unrelated CF patients are shown in A. HLA genes in A, are represented as red triangles, and x-axis represent standard and y-axis alternative mapping protocols. Panels B and C show gene expression quantifications by standard (x-axis) and alternative (y-axis) protocols in the format of FPKM for *HLA-DQB1*, and *HLA-DRB1* genes. Each dot represents 1 sample (out of 132 total), with solid line denoting linear regression line, and dashed line representing equality.

*CYP21A2* and HLA Class II genes at chr6 (S10 Fig in S4 File). The locus on chr16 (Fig 3, S5 Fig in S4 File) was borderline genome-wide significant that did not pass the threshold in publication of the GWAS study [1]. However, the chr16 region contains several genes relevant to CF lung disease, including *ERN2* involved in ER stress response and mucin production [83], and the *SCNN1B* and *SCNN1G* subunits of the epithelial sodium channel (ENaC) that have been suggested as being CF disease modifiers [84]. Over-expression of ENaC channels in *SCNN1B* transgenic mice has been used as a model of CF lung disease [85], and suppression of ENaC subunit expression is being explored as therapeutic strategies [86]. However, only *CHP2* and *PRKCB* in the chr16 region are consistently associated with CF lung disease by expression imputation (Figs 2 and 3, and Table 1).

Relevance to CF pathogenesis for the candidate modifiers are partly referenced in Table 1, and the full list of the 379 candidate genes often represent functional categories that are represented at the GWAS significant loci, for example *PDHX* discussed above (Table 2). Thus, both GWAS loci and non-GWAS loci contain genes that mark functions important in the pathogenesis of CF lung disease, such as immunity/infection/inflammation, virus/viral, and mucociliary clearance; and in CFTR biology, such as cytoskeleton, microtubules, mitochondria, lipid, ubiquitination, and ER and Golgi compartments. Several genes not in GWAS loci, e. g. *BPIFA1* [87–90], *CEACAM6* [91, 92], and *ORMDL3* [93–97], have been implicated directly in CF pathogenesis. Additionally, 4 genes (*RAC2*, *SDHA*, *TARS2*, and *YTHDF2*) have been

reported to be part of core *CFTR* interactome [98], so their mechanism of disease modification may partly be attributable to *CFTR* biogenesis. Another 6 genes (*AGER*, *ELAVL2*, *HLA-DQB1*, *JAZF1*, *MET*, and *RASSF3*) have recently been identified near genetic variants associated with lung function in COPD [99]. Interestingly, 11 genes are among the literature-curated transcription factors (Table 2), which are potential targets for intervention. Among them, *FOXP2* together with nucleotide binding protein, *NUBP1*, have been implicated in distal lung development in mice [100, 101], and the *NKX2-1*/*FOXP2* positive progenitor cells can be differentiated into distal alveolar cells [102]. These functional categories are also highly represented in GSEA analyses, with >60% of all enriched GSEA pathways representing these functional categories (S1, S2 Tables in S4 File). Further, highly similar pathways were observed in previous gene expression association studies [11, 12]. Taken together, these gene expression imputation results are congruent with current concepts of the pathophysiology of CF lung disease. All evidence of pathogenic relevance supports the validity of our data mining approach to uncover new genetic modifier genes of CF lung disease severity.

Among the 379 potential (and 52 consensus) modifiers, 92 (and 10) are non-protein-coding genes (S1 File and Table 1). There has been a rapid increase in identification of non-coding genes in recent years, with the current human genome assembly containing 20,433 protein-coding genes, 17,835 non-coding genes, and 15,952 pseudogenes (https://www.ncbi.nlm.nih.gov/genome/annotation_euk/Homo_sapiens/108/#FeatureCountsStats). There is little doubt that non-coding genes play important roles in biological functions, particularly in gene expression regulation [103–105], and evidence for their roles in CF disease processes are also emerging [106, 107]. The non-coding CF modifier genes reported here are likely under-estimated compared to protein-coding genes, due to reference genome and gene annotations associated with some of the gene expression data sets used in predictive model training, and general lag of functional knowledge of non-coding transcripts [108]. These are expected to improve over time, and new technologies and studies are required to understand mechanisms of CF disease modification by non-coding genes.

Although our efforts uncovered hundreds of potential candidate modifier genes from the CF GWAS data, it is likely not the whole story of genetic modification of CF lung disease severity, due to limitations of the data and necessary simplifications. The GWAS study with imputation can only effectively interrogate common variants, mostly SNPs, and gene expression imputation is currently restricted to autosomal genes due to the complexity of X chromosome gene expression between male and female samples, and apparent random selection of X-inactivation in females [109], thus, the GWAS signal for lung function on the X-chromosome [1] has not been interrogated. Furthermore, only *cis*-SNPs within 1 Mb (PrediXcan), or 0.5 Mb (TWAS) around a gene were used in predictive models of gene expression, and the genetic regulation of gene expression was modeled as linear additive effects of potential *cis*-SNPs. Therefore, modifier genes affected by rare variants were not investigated, and *trans*-regulation of gene expression was not evaluated. Additionally, some *cis*-regulation of gene expression may not follow linear combination (e.g. significant interaction between *cis*-SNPs), which would not be accurately assessed by current predictive models. Furthermore, the number of genes whose expression can be reliably predicted from genetic variants varied among tissues, ranging from ~2,000 to ~10,000, which in large part can be attributed to training sample sizes [10] (S2 Fig in S4 File). With continued accumulation of tissue samples and improved data quality, e. g. from GTEx, as well as improvement of gene expression quantification, and machine learning techniques, we expect to discover more candidate modifier genes of CF lung disease, and other CF related traits. To estimate proportion of genetic influences on CF lung disease phenotype from GWAS and gene expression imputation, we calculated heritability ($h^2$) from the imputed GWAS data using the GREML-LDMS method [19] from the Genome-wide Complex Trait

Analysis (GCTA) software [20]. The $h^2$ of KNoRMA from GWAS imputation of ~8.3 million SNPs among ~5,000+ unrelated CF patients, is 0.41 (SE = 0.072), while that from ~1.4 million *cis*-SNPs used in combined PrediXcan predictive models from 48 GTEx tissues, is 0.33 (SE = 0.061). The difference between the $h^2$ could potentially reflect missing imputable genes due to small training sample sizes, trans-regulation of gene expression from distant genetic variants, and/or other ways of affecting gene function from genetic variants.

The prevailing method of gene expression quantification used in published studies [5, 8, 10, 13] involved mapping of RNA-seq reads to the reference genome/transcriptome assembly, which are biased towards the reference sequences or alleles [82, 110]. This bias is more pronounced for hypervariable genes, such as some HLA genes, containing thousands of allotypes among the general population. When comparing alternative mapping strategies correcting for known variances and including multiple genome assemblies to the commonly used method (S13 Fig in S4 File), some genes (*HLA-DQA1*, *HLA-DRB1*) can change direction of association to CF lung disease from imputed gene expression, even though overall disease association are correlated (Fig 7) among the commonly imputable genes, as described [81, 82]. This indicates that reassessment of gene expression estimates based on HLA alleles in subset of samples can alter the predictive models, and subsequent association of imputed expression to disease phenotype in rare instances. However, the impact of allele-bias correction may be far reaching in that significantly more genes were imputed by SNP variants when RNA-seq reads were mapped with bias correction from our nasal epithelial biopsy data set (S2, S3 Files). This impact should be investigated with more data sets to understand genetic regulation of true gene expression.

In summary, we applied the technique of gene expression imputation, leveraging availability of CF and other eQTL data sets, to mine the CF GWAS data, and uncovered 52 consensus modifier genes for CF lung disease, which is substantially greater than identified by GWAS alone. Further, we identified an additional 327 potential candidate CF lung disease modifier genes. Some modifier candidates had been supported by independent studies, and functional annotations are consistent with our current knowledge of CF lung disease pathogenesis. These candidate modifiers provide potential targets for intervention of disease process in CF and for other airway diseases as well.

## Supporting information

**S1 File.**
(XLSX)

**S2 File.**
(XLSX)

**S3 File.**
(XLSX)

**S4 File.**
(DOCX)

## Acknowledgments

We thank Dr. Nancy J. Cox, Vanderbilt University, Division of Genetic Medicine, Dr. Fred Wright, North Carolina State University, Bioinformatics Research Center, and Dr. Ani W. Manichaikul, University of Virginia, Center for Public Health Genomics, for guidance, advisement, and discussion. We also like to thank Dr. Hae Kyung Im and lab, University of Chicago,

Department of Human Genetics, Dr. Alexander Gusev and lab, Harvard University, Dana Farber Cancer Institute, and the Genotype-Tissue Expression (GTEx) project, for making their software tools and databases (PrediXcan and TWAS) open source and publicly available.

## Author Contributions

**Conceptualization:** Hong Dang, Wanda K. O'Neal, Michael R. Knowles.

**Data curation:** Hong Dang, Deepika Polineni, Harriet Corvol, Garry R. Cutting, Mitchell L. Drumm, Lisa J. Strug, Michael R. Knowles.

**Formal analysis:** Hong Dang.

**Funding acquisition:** Michael R. Knowles.

**Investigation:** Deepika Polineni.

**Methodology:** Hong Dang, Wanda K. O'Neal.

**Project administration:** Rhonda G. Pace.

**Resources:** Wanda K. O'Neal, Michael R. Knowles.

**Software:** Hong Dang.

**Supervision:** Wanda K. O'Neal, Michael R. Knowles.

**Visualization:** Hong Dang.

**Writing – original draft:** Hong Dang.

**Writing – review & editing:** Rhonda G. Pace, Jaclyn R. Stonebraker, Garry R. Cutting, Lisa J. Strug, Wanda K. O'Neal, Michael R. Knowles.

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
