## [Decision Letter · Decision Letter 0]

1 Nov 2019

PONE-D-19-23859

Mining GWAS and eQTL data for CF lung disease modifiers by gene expression imputation

PLOS ONE

Dear Dr. Dang,

Thank you for submitting your manuscript to PLOS ONE. After careful consideration, we feel that it has merit but does not fully meet PLOS ONE’s publication criteria as it currently stands. Therefore, we invite you to submit a revised version of the manuscript that addresses the points raised during the review process.

The manuscript provides a comprehensive transcriptome-wide association study that identifies a number of genes which may modify CF severity, greatly increasing knowledge of CF genetics. The reviewers have raised a number of issues which should be addressed, particularly in the context of the statistical analysis, and additional information is required to improve the clarity and comprehensibility of the study.

We would appreciate receiving your revised manuscript by Dec 16 2019 11:59PM. To enhance the reproducibility of your results, we recommend that if applicable you deposit your laboratory protocols in protocols.io, where a protocol can be assigned its own identifier (DOI) such that it can be cited independently in the future. For instructions see: http://journals.plos.org/plosone/s/submission-guidelines#loc-laboratory-protocols

We look forward to receiving your revised manuscript.

Kind regards,

Dylan Glubb

Academic Editor

PLOS ONE

Journal Requirements:

Reviewers' comments:

Reviewer's Responses to Questions

**Comments to the Author**

1. Is the manuscript technically sound, and do the data support the conclusions?

Reviewer #1: Yes

Reviewer #2: Yes

Reviewer #3: Partly

2. Has the statistical analysis been performed appropriately and rigorously? 

Reviewer #1: Yes

Reviewer #2: Yes

Reviewer #3: No

3. Have the authors made all data underlying the findings in their manuscript fully available?

Reviewer #1: Yes

Reviewer #2: Yes

Reviewer #3: No

4. Is the manuscript presented in an intelligible fashion and written in standard English?

Reviewer #1: Yes

Reviewer #2: Yes

Reviewer #3: No

5. Review Comments to the Author

Reviewer #1: The authors describe an extension of a previous GWAS concerning cystic fibrosis (CF) by imputing gene expression with PrediXcan and TWAS using GTEx, LCL, and nasal mucosal models. They take the genes at the consensus of the most significant results of both methods and examine how they function within the CF pathway as well as delve into genes not near known loci and how their genetic regulation contributes to CF. These analyses provide valuable insight into the genetic mechanisms underlying CF in addition to finding potential therapeutic targets.

While your reasoning behind your methods was sound, the interpretation of PrediXcan and TWAS’ results and the decision to take the consensus of the methods is my main concern. Additionally, as an overall note, please use color in your grey and black plots while possible, it makes it much easier to discern the data.

Thank you for making GWAS summary statistics available for public. Are your PrediXcan models from the CF and nasal mucosal epithelial cells freely available as well?

36-37: “Using congruence of findings from the two approaches” - at this point in the abstract, you haven’t compared the two approaches

73-75: “Thus, using eQTLs from all tissues, not just “disease-relevant” tissues, can identify a greater number of candidate gene modifiers” - can you elaborate more on how genes in non-relevant tissues may be contributing to CF?

104-105: “The cohort study design, and demographic and clinical characteristics of the CF patients used in this study have been previously described” - a quick summary of the cohort (sample size, number of SNPs, sex, age, race/ethnicity composition), etc. would be welcome as an addition to the paragraph.

118: “more recent release of 1000 genomes project data” - specify which phase

120-122: “summary GWAS statistics… derive imputed gene expression… using [FUSION]” - an issue with comparing PrediXcan and FUSION output is their inherent differences in design. For more comparable results if you continue using both methods, you should consider using S-PrediXcan (29739930) which is similar to PrediXcan but takes summary statistics rather than genotypes as input. In addition, description of your methods (ex. Models used) in FUSION to the same extent as PrediXcan would be appreciated.

131: “the samples used in predictive model training were excluded from the association testing” - what’s the before and after n?

137-138: “In our most stringent analyses, we sought consensus between two meta-analysis approaches.” - The reason behind taking the intersection of these two methods, rather than just the results from one model (especially since only GTEx is used in FUSION while there are additional models for PrediXcan), isn’t articulated well through the paper. Either take the results from the more robust method, or fully justify the use of the intersection of the methods’ results.

141: “performed for each gene among the tissues with imputed gene expression” - how does this account fairly between genes ex. if one of them is present in only two tissues and the other is in thirty related tissues?

183-184: “our LCL microarray data set yielded fewer than expected number of imputable genes” - what is the expected # of imputable genes, based on PrediXcan models produced with microarray data? Additionally, in the following paragraph, how is the difference between microarray and RNA-Seq in CF LCL vs. GTEx LCL accounted for in this comparison?

200-202: “Using a threshold of p-value < 0.01… disease modifier genes were identified.” - Does this method account for the range of tissues used? Does this account for the CF and DGN models not existing in TWAS? Can you delve into how a result in a seemingly irrelevant tissue can still be useful in elucidating CF pathways? These questions especially arise in lines 211-214.

206-208: “average effect sizes… (R2 = 0.36, Fig 3B)” - these calculations should be covered in Methods

240-243: “MET ~700 kb upstream… in predictive models.” - How would this compare to a traditional GWAS-eQTL analysis?

259-261: “The SNP p-values used in this analysis were either the minimal p-value selected per gene (Fig. 5) or represented the average of the unique set of SNPs from all predictive models per genes (S6 Fig).” - Choose a consistent metric across all genes.

282: “among SNPs with significant p-values of < 10e-7” - why did you use this threshold instead of the most traditional 5e-8?

304-305: “largely due to environmental influences and/or disease process, rather than genetic regulation.” - do you have an h2 measurement for this (GWAS vs. imputed gene expression)? This section would also flow better in the creation/analysis of the models near the beginning of results.

315: “maximal p-value between the 2 multi-tissue meta-analyses for each analysis platform” - Again, choose a consistent metric from one platform or else results are difficult to interpret.

360-361: “the direction of predicted expression changes in regard to lung function are opposite between different mapping strategies” - which method best agrees with the known direction of effect in observed CF gene expression data?

392-394: “Although EHF and APIP… current gene expression data.” - this issue of best predicted in a model vs. the genes with actual biological implications was described in Weinberg et al. (PMID: 30926968). How does this perspective, as well as the differences in the tissues, contribute to PDHX as a candidate in the CF pathway? How would a colocalization analysis change these results? Additionally, what is the gene currently implicated in the locus in that may be related to CF?

399-410: Good review of how the most sig. predicted genes could work in the context of CF!

433-434: “The overall gene expression association to the CF lung disease severity from our own CF nasal epithelial and LCL data sets is not correlated with imputed gene expression.” - this seems contradictory to lines 175-178, can you clarify this?

444: “The non-coding CF modifier genes are likely underestimated” - in addition, they seem understudied from the fewer references in table 1 compared to the protein coding genes. Can you elaborate on the reasons why this may be?

460-461: “Furthermore, the number of genes whose expression can be reliably predicted from genetic variants varied among tissues” - a noticeable absence throughout most of this paper is the lack of individual tissues being scrutinized. Which tissue are these significant genes in, and how may that also contribute to CF? And if the tissue seems irrelevant, why are the findings still important?

Next, review figures/tables and supplementals

Table 1: add a column of which tissue the P-value was max. Where did the keywords originate from? Can you use coefficients instead of protective/harmful?l

Fig. 1: Since the PM, imGE, and pvals for GTEx, DGN, nasal epi, and LCL are similar except for sample sizes, can they be consolidated to look less cluttered? Also, since genotypes were used to make the gene expression models for nasal epi and LCL and not summary statistics, “GWAS imputation” may be misleading.

Fig. 2: There’s a lot going on in this plot that makes it difficult to follow. Is it more legible if you subset to only the most important tissues?

Fig. 3: Use ggrepel with higher force for easier readability. These plots would be more appropriate in the supplemental figures with the r2 of both of them described in the text.

Fig. 4: At a glance, the PrediXcan and TWAS results look similar in architecture. Can these results be consolidated with colors identifying differences between the methods instead, or colors indicating known vs novel CF genes?

Fig. 5: This figure would also benefit from ggrepel. It would be easier to follow as a 2x2 rather than branching off the original scatterplot. Are all 3 of these offshoots necessary? What do each of them represent that the others don’t?

Fig. 6: Very unique and insightful visual. Can you discuss more about how these genes are co-regulated from their intertwined, linked eQTLs?

Fig. 7: Put the r2 in captions. This figure would also be better suited for a supplemental.

Fig. 8: Can you differentiate the known genes from the novel ones with color, bolding, or other formatting? Do you have similar figures for “consensus” genes available?

Fig. 9: Color in the HLA gene dots, they get lost amongst all the grey.

Supporting methods: I enjoy your well-described and well-cited (with links!) descriptions. There are a few very minor typos - “the imputed gene expression data sets had sample size” should be “the imputed gene expression data sets had a sample size”. I have a small concern with the hierarchical clustering analysis. Why did you have “the missing values in the resultant distance matrix were replaced with the largest distance values between any pairs” rather than just leaving the data N/A? It seems misleading. Also, again, for a PrediXcan-like method that also uses summary statistics, S-PrediXcan is up your alley.

S1 Fig.: Do you think these dramatically different slopes are due to RNA quantification collection methods, sample sizes, or other outside factors?

S2 Fig.: I quite enjoy this figure. Can you compare the LCL microarray observed/expected gene count to those found in PrediXcan microarray-based models?

S6 Fig.: This is illegible. Would you be able to give each point a number and then have a side table with both the numbers and gene names?

S11 Fig.: Can you include within the main text as part of the discussion why CFTR isn’t as highly significant as one would initially think in an analysis of cystic fibrosis?

S Tables: Can you bold or italicize the known or novel gene findings to make them easier to differentiate?

Overall, my concerns lie mainly with the comparison between PrediXcan and TWAS, the interpretability of the figures, and the lack of connecting the actual tissue genes were determined significant to the CF phenotype, but I enjoy your analysis and your contribution to determining the genetic architecture of CF. Addressing these concerns as well as clarifying the points I made above and adding an additional colocalization analysis would strengthen this paper.

Reviewer #2: This is a very comprehensive and well-conducted analysis.

I have two questions:

1. The harmonic mean P-value method was recently corrected ( please see: http://blog.danielwilson.me.uk/2019/08/updated-correction-harmonic-mean-p.html ). Do the authors' calculations incorporate the updated (and corrected) harmonic mean P-value method?

2. There is no discussion of LD-contamination ( see for example: https://www.nature.com/articles/s41467-018-03621-1 ). To what extent does this affect the results?

Reviewer #3: On the whole, the writing of the manuscript can and should be improved. As it stands, it is hard to follow. The authors present no motivation for the selected approaches to analyze the data (and why they chose to use two). The methods are not described well. It takes a great effort to match the description of the analyses with Fig. 1 illustrating them.

Description of one type of analysis is interweaved with sentences starting with ‘Alternatively’ (lines 120, 132) describing the other type of analysis. This makes it harder to follow either of them.

The authors pay attention to unimportant details (such as talk about GTEx pilot data) but not discuss important ones (e.g. choice of just 1 principal component to correct for ancestry).

It causes great concern to see that the authors did not correctly use scientific E-notation for small numbers.

For instance, instead of ‘1e-6’ or ‘10^{-6}’ the authors have 1x10e-06, which would actually be equal to 1e-5 if read correctly (10x1e-6 = 1e-5).

There are a total of seven instances of incorrect use of scientific E-notation.

Which samples were whole genome sequenced (WGS)?

Is the sequencing data publicly available?

Were these samples among those being imputed?

Were WGS genotypes mixed with imputed genotypes in the analyses?

Line 127. Why only one PC was used for correction of ancestry? This appears to be grossly insufficient given current knowledge.

Line 129. There are numerous methods for robust regression analysis. Not providing a name for the chosen method, only a citation is a great inconvenience for the reader.

Lines 137-138. What does “in our most stringent analyses” mean?

Line 138. The phrase “we sought consensus” does not exactly read as “we selected genes significant in both analyses”.

Line 173. What is the definition of 'imputable gene'?

Minor comments:

Line 72. Why even mention GTEx pilot data?

Why every plot is black and white?

6. PLOS authors have the option to publish the peer review history of their article (what does this mean?). If published, this will include your full peer review and any attached files.

Reviewer #1: No

Reviewer #2: No

Reviewer #3: No

---

## [Author Response · Author response to Decision Letter 0]

1 Sep 2020

NOTE: Responses to reviewers with proper formatting are attached as a .docx file, and only the text is copied below.

I. General Overview of Responses

We appreciate the effort and time the reviewers have put into the reviews. We have taken their suggestions very seriously, and we believe that responding to their comments has significantly improved the manuscript. The responses to each of the reviewer comments is provided below. As way of introduction, there were four main points that we thought were the thrust of the reviewer’s concerns. We summarize our overall response to these four concerns here, while providing detailed response notes later in the document.

1. The reviewers were concerned that the analyses between PrediXcan and TWAS are not uniform, and so it is not possible for one to “replicate the other”. Thus, they were unsure whether our concept to provide what we previously called our “most robust” list of candidate genes was valid. We understand their confusion and apologize that our overall intent for choosing to report on the two approaches was not clear. It is NOT our intent to compare and contrast the two similar but independently developed platforms of gene expression imputation (PrediXcan and TWAS), or to ask one to replicate the other. Rather we wish to mine the data to obtain maximal coverage. We believe our goal was best achieved by utilizing these two divergent methods, leveraging the strength of each method. Importantly, intersecting gene modifier candidate signals are generated by both methods. To illustrate the value of the analyses for the CF field, we chose to focus on the consensus findings between the two approaches. We believe for the purposes of this paper that this “consensus” list represents the most robust results, as they are stable between the two approaches. Since the candidate modifier genes extracted from the data mining constitute hypotheses that need to be validated by independent experiments, which requires time and resources, we sought to evaluate the main findings by reviewing functional annotations and literature as indirect validation in the context of CF biology. This “consensus” list is a workable list that provided a framework for our discussion. We have modified the discussion of the data mining substantially and hope that our new explanation provides clarity. 

2. The reviewers were concerned about the number of tissues that were explored in our analyses. It is true that our method utilized multiple tissue eQTL data, which relies on many tissues that are not known to be directly affected in CF disease pathogenesis. We chose this strategy based on the findings by GTEx that the majority of genetic regulation of gene expression is through cis-SNPs, which tend to be conserved across multiple tissues. Under the concepts developed by GTEx, extracting predictive relationships of SNPs to gene expression from all available tissues helps to overcome technical issues of uneven sample number, data quality among the different tissues, biological influences of environments and development, or reverse causality (disease affecting gene expression). We cite several papers that support this concept.

3. The reviewers were concerned that only one PC was used to control for population stratification. We appreciate this as a legitimate concern and revised the analysis, now including 4 PCs. This did reduce the number of candidate disease modifier genes slightly, and so is a more conservative approach. Please note however that the top candidate genes remain, and the overall concepts related to the biology of CF modifier-related pathways was not dramatically altered by this change.

4. The reviewers were also concerned that issues related to LD structure and causal SNPs for disease modification were overinterpreted. We agree with the sentiment that the data mining approach and findings do not constitute proof of causality or identify causal SNPs. We acknowledge that strong LD between SNPs, while not impeding the accuracy of gene expression prediction, presents considerable challenge for causality determination. Nonetheless, by highlighting genetic regulation of gene expression from available eQTL data, i.e., pointing out genes with or without demonstrable eQTLs, the analysis we present puts us a step closer to the mechanism of disease modification. Future studies to test hypothesis related to the significant findings will be needed to link gene expression to lung phenotype.

II. Specific Reviewer Comments and Responses

Reviewer 1

Overview: The authors describe an extension of a previous GWAS concerning cystic fibrosis (CF) by imputing gene expression with PrediXcan and TWAS using GTEx, LCL, and nasal mucosal models. They take the genes at the consensus of the most significant results of both methods and examine how they function within the CF pathway as well as delve into genes not near known loci and how their genetic regulation contributes to CF. These analyses provide valuable insight into the genetic mechanisms underlying CF in addition to finding potential therapeutic targets.

While your reasoning behind your methods was sound, the interpretation of PrediXcan and TWAS’ results and the decision to take the consensus of the methods is my main concern. Additionally, as an overall note, please use color in your grey and black plots while possible, it makes it much easier to discern the data.

Response to Overview: We have attempted to clarify our line of thinking as described in the overview above. In general, our intent was to conduct the analyses using the divergent approaches (Predixcan and TWAS), which complement each other and have individual strengths, to maximize discovery potential for hypothesis generation. We then sought to evaluate the significance of the findings through biological context. Utilizing the consensus of findings between the two different approaches was used as a prioritization tool, allowing us to focus the biological exploration on the most robust findings. 

Please note that all findings will be provided from both methods. 

As a note, we have incorporated color into the plots where it is helpful.

Reviewer 1, Comment 1. Thank you for making GWAS summary statistics available for public. Are your PrediXcan models from the CF and nasal mucosal epithelial cells freely available as well?

Response: Yes, we will make these models available. We are planning on hosting these files on GitHub, unless the journal has a more preferred option, which we will explore with them. The hosted data will include a GWAS summary, PrediXcan models, individual tissue results, combined meta-analysis results, pathway results, and data values used to generate figures.

Reviewer 1, Comment 2. 36-37: “Using congruence of findings from the two approaches” - at this point in the abstract, you haven’t compared the two approaches

Response: We have modified the text (lines 34-37): “By comparing and combining results from alternative approaches, we identified 379 candidate modifier genes. We delved into 52 modifier candidates that showed consensus between approaches, and 28 of them were near known GWAS loci”

Reviewer 1, Comment 3. 73-75: “Thus, using eQTLs from all tissues, not just “disease-relevant” tissues, can identify a greater number of candidate gene modifiers” - can you elaborate more on how genes in non-relevant tissues may be contributing to CF?

Response: As discussed above, the concept is not that genes showing signal in non-relevant tissues are only active in those non-relevant tissues, but that the signals derived from these tissues add to the power to detect signals. As discussed in the survey of early GTEx data sets (1, 2) (references at the end of this document), most genetic regulations of gene expression are local through cis-SNPs, and many of them are shared across multiple tissues. So by looking at patterns of gene expression regulation by cis-SNPs from all tissues, one can recover more trait modifier genes, presumably by overcoming non-optimal training data, e.g., small sample sizes, and potential biological limitations, e.g., temporal regulation during development and potential effects of disease progression on gene expression in affected tissues. 

We hope that we have clarified this in the revised text (lines 68-72): “In other words, genetic regulation of gene expression, or eQTL, can be informative regardless of tissue origin of the training data set (2), and can help overcome technical deficiencies, such as small sample sizes of certain tissue data, and potential biological limitations such as unsampled developmental stage and environmental and pathogenic masking of gene expression through reverse causality.”

Reviewer 1, Comment 4. 104-105: “The cohort study design, and demographic and clinical characteristics of the CF patients used in this study have been previously described” - a quick summary of the cohort (sample size, number of SNPs, sex, age, race/ethnicity composition), etc. would be welcome as an addition to the paragraph.

Response: We have added summary description of the CF GWAS cohorts as follows: “Briefly, 5 cohorts (total 6,365 CF patients) with >90% European ancestry from US, Canada, and France were recruited by the International Cystic Fibrosis Gene Modifier Consortium, and their genome-wide genetic variance were assayed using different genotyping platforms over several years. GWAS was performed as a meta-analysis of cohort/platform combinations, using the standardized lung function score, KNoRMA, as phenotype trait (3, 4).”

Reviewer 1, Comment 5. 118: “more recent release of 1000 genomes project data” - specify which phase

Response: The exact version is “1 KG phase3 v5a haplotype” data (5). Revised text (lines 132-134): “Compared to the imputation reported in the GWAS studies (3), the updated version here utilized a more recent release of 1000 genomes project Phase3 (v5a) haplotype data and 101 CF whole genome sequencing data as reference panels, which improved coverages at HLA and CFTR regions (5).”

Reviewer 1, Comment 6. 120-122: “summary GWAS statistics… derive imputed gene expression… using [FUSION]” - an issue with comparing PrediXcan and FUSION output is their inherent differences in design. For more comparable results if you continue using both methods, you should consider using S-PrediXcan (29739930) which is similar to PrediXcan but takes summary statistics rather than genotypes as input. In addition, description of your methods (ex. Models used) in FUSION to the same extent as PrediXcan would be appreciated.

Response: We have added more description of the FUSION procedure in the main text. In our original version, we made efforts to simplify the description of methods, provided more detail in the Supporting Methods section, but perhaps we went overboard with that approach and too many details were missing. We have sought to remedy this by trying to strike a better balance between what we present in the main text versus what we have detailed in the Supporting Methods. In response to this specific comment by the reviewer, we have added more details to the main text.

The new text added (lines 149-153): “Briefly, summary GWAS statistics for SNP associations to CF lung disease phenotype (n=6,365) and reference linkage-disequilibrium (LD) data from 1000 genome projects were used as input for FUSION, with TWAS predictive models from 48 GTEx v7 human tissues downloaded from FUSION website (http://gusevlab.org/projects/fusion/). The analysis was performed according to instructions on the FUSION website.”

As we hoped we have clarified, our goal is not to compare different approaches, but rather to mine the data, and by using a consensus of findings between the different approaches to generate a list of the consensus results, thus, maximizing discovery and hypothesis generation. That being said, we did explore S-PrediXcan (MetaXcan) as a potential analyses option. As shown in the following figure, the meta-analysis p-values by Harmonic Mean Pvalue (HMP) from 48 GTEx tissues are highly correlated pair-wise between PrediXcan, MetaXcan, and TWAS (the 99 consistent HMP p-value<0.01 significant genes among all 3 sets are highlighted).

As can be appreciated, the majority of the genes detected by S-PrediXcan (MetaXcan) were also detected by our original methods (TWAS and PrediXcan). Since there is no readily available meta-analysis for dependent multiple p-values other than HMP, and there are fewer significant genes (p<0.01) for MetaXcan compared to PrediXcan (346 vs 478) from the same tissues, we chose to stay with our original comparison of PrediXcan and TWAS.

Reviewer 1, Comment 7. 131: “the samples used in predictive model training were excluded from the association testing” - what’s the before and after n?

Response: We have added the relevant text to clarify this issue. The total unrelated sample size is 5,756 (selecting one out of siblings from the same family following rules set out by the GWAS analysis). There are 122 and 753 samples among the 5,756 in the nasal epithelial and LCL training data sets, respectively. Therefore, excluding them from lung disease association tests resulted in 5,634 and 5,003 final sample sizes for nasal epithelial and LCL tissues, respectively.

Revised text (lines 138-141, 143-146): “Association testing of imputed gene expression, using the PrediXcan platform (6), from the CF LCLs and nasal epithelium, 48 GTEx tissues, and DGN whole blood (a total of 51 human tissues), were performed using robust regression (7, 8) based on 5,756 unrelated patients .”… “For disease phenotype association testing using predictive models trained on CF nasal epithelial and LCL data sets, the samples used in predictive model training were excluded from the association testing, resulting in 5,634 and 5,003 final sample size for nasal epithelial and LCLs, respectively.”

Reviewer 1, Comment 8. 137-138: “In our most stringent analyses, we sought consensus between two meta-analysis approaches.” - The reason behind taking the intersection of these two methods, rather than just the results from one model (especially since only GTEx is used in FUSION while there are additional models for PrediXcan), isn’t articulated well through the paper. Either take the results from the more robust method, or fully justify the use of the intersection of the methods’ results.

Response: We hope that we have now fully justified the use of the intersection of the methods in our response. As discussed above, we think the exploratory nature of our analyses justifies our strategy to utilize two independently developed approaches – each with different strengths and weaknesses. Correlation of effect size estimates and consensus of significant genes between the 2 result sets help strengthen the main findings and identify robust candidates for follow-up as modifier genes. The union of results maximizes coverage of potential modifiers, which is important at current state of rapid development, where many things, such as training data sample sizes and quality of gene expression estimates are not optimal. In fact, we were comforted by the fact that some of the expanded list of candidate modifier genes, such as BPIFA1, which has been strongly implicated to modify CF lung disease in the literature, seemed to help justify our nuanced strategy.

Reviewer 1, Comment 9. 141: “performed for each gene among the tissues with imputed gene expression” - how does this account fairly between genes ex. if one of them is present in only two tissues and the other is in thirty related tissues?

Response: We have investigated the relationship between meta-analysis p-value and the number of tissues a gene was imputed by examining the combined meta-analysis p-values for the 52 consensus modifier genes. As shown by the intensity patterns in Figure 2 of our manuscript, significant phenotype association of imputed gene expression appears to be determined by at least 2 factors, close distance to GWAS loci, and number of tissues the gene is imputed. This makes sense since the imputed gene expression reflects the underlying genetic variances. On the other hand, genes can be significant due to strong association in a few imputable tissues, or consistent weak associations in many imputed tissues through meta-analysis. We believe, although significant genes imputed from multiple tissues should be given higher priority for follow-up studies, the genes imputed in only a few tissues may still be important since they can be tissue-specific, or insufficiently sampled due to non-optimal sample sizes in the reference tissue data sets.

Reviewer 1, Comment 10. 183-184: “our LCL microarray data set yielded fewer than expected number of imputable genes” - what is the expected # of imputable genes, based on PrediXcan models produced with microarray data? Additionally, in the following paragraph, how is the difference between microarray and RNA-Seq in CF LCL vs. GTEx LCL accounted for in this comparison?

Response: All PrediXcan models used in this report were based on RNA-seq gene expression assays, with the exception of our CF LCL data set. While one data set makes it difficult to estimate expected number of imputable genes from microarray assays, our LCL data point deviates significantly from the gene count vs sample size relationship established by many tissue training data sets based on RNA-seq (figure below, left panel). Similarly, Gusev et al also demonstrated the reduced number of imputable genes from the Netherland Twin Registry (NTR) microarray gene expression data (9), Figure 3 in their paper, and reproduced below. It is difficult to compare number of imputable genes from our LCL (microarray, 5,299 genes), and GTEx LCL (RNAseq, 3,023 genes), since the sample numbers are quite different: 753 CF LCL vs 117 GTEx LCL, and the correlation of the 1,623 common imputed genes between the 2 sets at imputed gene expression of ~5,000 patients did not take original assay platform into consideration. Our assumption is gene expression data quality affects predictability at model training stage, and the number of genes that can be predicted from cis-SNPs given the same sample size, so it was accounted for at model training.

Reviewer 1, Comment 11. 200-202: “Using a threshold of p-value < 0.01… disease modifier genes were identified.” - Does this method account for the range of tissues used? Does this account for the CF and DGN models not existing in TWAS? Can you delve into how a result in a seemingly irrelevant tissue can still be useful in elucidating CF pathways? These questions especially arise in lines 211-214.

Response: As stated above in response to Comment 3, there are both technical and biological rational to leverage predictive models of genetic regulation of gene expression from all tissues where reference data are available. We also intended to use PrediXcan and TWAS as complementary approaches to mine the CF GWAS data, not to compare the merits between them, therefore, we feel the fact that DGN and CF models are not available for TWAS analysis does not adversely affect the intended use.

Reviewer 1, Comment 12. 206-208: “average effect sizes… (R2 = 0.36, Fig 3B)” - these calculations should be covered in Methods

Response: We have now added a section to describe the comparison of results by linear regression in Supporting Methods. The revised text (lines 146-148): “To assess correlation between different test results among multiple genes, simple linear regression was performed between 2 sets of test statistics, such as mean effect-sizes, or -log10 p-values from PrediXcan and TWAS meta-analyses, or GWAS.”

Reviewer 1, Comment 13. 240-243: “MET ~700 kb upstream… in predictive models.” - How would this compare to a traditional GWAS-eQTL analysis?

Response: We have not specifically compared the predictive models derived from machine learning (i.e. panelized regressions) with traditional eQTL, but the original PrediXcan and TWAS papers did address the issue generally, and they found that the machine learning approach is more accurate than single variant – gene eQTL analysis (6, 9) for predicting gene expression. Intuitively, predictive models with multiple SNPs as independent variables would include single SNP eQTL as a special case, if the eQTL is strong enough to explain the observed gene expression variance. As shown in Supplemental Figure S11, the MET signal came from multiple sub-threshold GWAS SNPs, which may also show up in traditional GWAS-eQTL overlap test, depending on the parameters.

Reviewer 1, Comment 14. 259-261: “The SNP p-values used in this analysis were either the minimal p-value selected per gene (Fig. 5) or represented the average of the unique set of SNPs from all predictive models per genes (S6 Fig).” - Choose a consistent metric across all genes.

Response: We are sorry for the confusion. The metrics are consistent across all genes – the different metrics refer to alternative comparisons as plotted in different figures, not among different genes. We hopefully have clarified this issue in the main text. We feel it is helpful to provide both minimal and mean p-values when comparing GWAS at SNP level to imputed expression at gene level, since typically many SNPs were used to predict gene expression. Therefore, genes can be ranked from GWAS by the most significant SNP (minimal p-value) in the model (Figure 4, formerly Figure 5), or mean p-value among all the SNPs in the model (updated Supp. Figure S6), and both showed significant correlation with imputed gene expression association to phenotype. 

Reviewer 1, Comment 15. 282: “among SNPs with significant p-values of < 10e-7” - why did you use this threshold instead of the most traditional 5e-8?

Response: Our goal for this part of the manuscript is to illustrate the connection between GWAS SNP associations to gene expression regulation by showing the inclusion of strongly associated SNPs in gene expression predictive models. This is somewhat arbitrary, since we are looking for SNPs associated with CF lung function that may also regulate gene expression, and it is not necessarily constrained by genome-wide significant threshold. We chose a highly suggestive significance level (<10-07) with the assumption that such an association to lung function is more likely to be reflected in the imputed gene expression, since genome-wide threshold of 5e-8 would result in fewer SNPs to overlap with those chosen in the predictive models. 

Reviewer 1, Comment 16. 304-305: “largely due to environmental influences and/or disease process, rather than genetic regulation.” - do you have an h2 measurement for this (GWAS vs. imputed gene expression)? This section would also flow better in the creation/analysis of the models near the beginning of results.

Response: After reconsidering the complex issue of genetic vs environmental influences on phenotype traits, and the potential confusion of “negative” results of no correlations as raised also by Reviewer 1, Comment 21, we have decided to drop this topic, and figure from our manuscript.

We did investigate the issue of heritability as suggested by the reviewer, and estimated h2 of CF lung function score (KNoRMA) to genome-wide SNPs, using the Genome-wide Complex Trait Analysis (GCTA) approach. The h2 from all imputed SNPs (~8.3 million) is 0.41, and that from cis-SNPs of all PrediXcan predictive models (~1.4 million SNPs) is 0.33. We interpret these preliminary findings as estimates of narrow-sense inheritance of common SNPs of CF lung function, and 0.41 represent the upper limits gene expression imputation can achieve. The h2 of 0.33 from all cis-SNPs used in current gene expression imputation suggests that there is still room for data mining of the GWAS signals.

We have added relevant text in discussion (lines 478-486): “To estimate proportion of genetic influences on CF lung disease phenotype from GWAS and gene expression imputation, we calculated heritability (h2) from the imputed GWAS data using the GREML-LDMS method (10) from the Genome-wide Complex Trait Analysis (GCTA) software (11). The h2 of KNoRMA from GWAS imputation of ~8.3 million SNPs among ~5,000+ unrelated CF patients, is 0.41 (SE = 0.072), while that from ~1.4 million cis-SNPs used in combined PrediXcan predictive models from 48 GTEx tissues, is 0.33 (SE = 0.061). The difference between the h2 could potentially reflect missing imputable genes due to small training sample sizes, trans-regulation of gene expression from distant genetic variants, and/or other ways of affecting gene function from genetic variants.” 

Reviewer 1, Comment 17. 315: “maximal p-value between the 2 multi-tissue meta-analyses for each analysis platform” - Again, choose a consistent metric from one platform or else results are difficult to interpret.

Response: We acknowledge the difference in general strategy we adopted and the desire for uniform metric across all analyses. Our past experiences dealing with high-throughput -omics data sets of 20k+ genes and millions of SNPs suggested that there are often spurious results and edge cases in such analysis, and we therefore sought agreement between different approaches to ensure robustness of the findings. The results tend to be more conservative as evaluated by the distribution of final p-values shown in the QQ plots below, which demonstrate effective control of Type I error. We believe functional evaluations and literature review of the resultant candidate CF lung disease modifier genes supported such a strategy.

Reviewer 1, Comment 18. 360-361: “the direction of predicted expression changes in regard to lung function are opposite between different mapping strategies” - which method best agrees with the known direction of effect in observed CF gene expression data?

Response: This is an interesting question! In our opinion, genetically regulated gene expression need not have the same direction of association to the phenotype trait as the actual observed gene expression, since observed gene expression can also be highly influenced by environmental factors and/or disease processes. Examining the current results for the 2 genes, HLA-DRB1 and HLA-DQA1, it happened to be the case that the alternative mapping strategy of accounting for extra allele polymorphisms at the HLA loci resulted in the same direction of lung disease association between imputed and observed gene expression, while the commonly employed mapping protocol resulted in the opposite direction. However, this is a complex issue that needs more careful investigation, which we feel is beyond the scope of this paper. The ultimate proof is to knock-out a modifier gene, and then examine its effect on the phenotype trait as predicted by this approach.

Reviewer 1, Comment 19. 392-394: “Although EHF and APIP… current gene expression data.” - this issue of best predicted in a model vs. the genes with actual biological implications was described in Weinberg et al. (PMID: 30926968). How does this perspective, as well as the differences in the tissues, contribute to PDHX as a candidate in the CF pathway? How would a colocalization analysis change these results? Additionally, what is the gene currently implicated in the locus in that may be related to CF?

Response: This is a fascinating topic, and it touches on a newly developed method, which we have considered and followed both the Wainberg paper (12) and the responses from the PrediXcan authors online: http://hakyimlab.org/post/2017/vulnerabilities/, in writing our discussion. As users of this method, and evaluating the results more from known CF biology, we resonate with opinions expressed by both the Wainberg paper and PrediXcan authors, in that this is an incremental progress, that gets us closer to the causal variant(s) and mechanism, but does not get all the way there. These methods have some useful properties of framing the analysis at gene expression level, which improves power identifying modifier genes outside the genome-wide significant loci, narrows the candidate gene list around GWAS loci by ignoring genes without eQTL support. Apart from algorithm improvement and best practices, an important issue is that our reference eQTL training data are far from perfect – small sample sizes (most <300), mixture of cell types from whole tissues, limited time representations lacking many developmental stages of tissues, all of which impede mechanistic discovery. Improvement in sample size and quality is probably more impactful, at this time, on the overall performance of these approaches. As a concrete example, we have been studying EHF and APIP at the chr11 GWAS locus for several years without major breakthrough, and this analysis points to genes further away from the peak SNPs by predictive models, which if validated by further investigation and prove to be true, will certainly advance our understanding of genetic modification through gene expression regulation. Meanwhile, EHF and APIP cannot be excluded due to lack of or little genetic regulation or eQTL signals, since it is possible actions of these genes in specific cells or developmental stages relevant to CF lung disease are not captured by current eQTL data sets. 

Reviewer 1, Comment 20. 399-410: Good review of how the most sig. predicted genes could work in the context of CF!

Response: Thank you. 

Reviewer 1, Comment 21. 433-434: “The overall gene expression association to the CF lung disease severity from our own CF nasal epithelial and LCL data sets is not correlated with imputed gene expression.” - this seems contradictory to lines 175-178, can you clarify this?

Response: Although we have dropped the content relating comparison of observed vs imputed gene expression associations to disease phenotype (Reviewer 1, comment 16), it is a good question we would like to respond. Among the genes whose expression can be imputed from cis-SNPs in nasal epithelial (2,881) and LCLs (5,299), most (2,309 and 4,633) had also been tested for disease phenotype association of the observed gene expressions, which overall are not correlated with those of imputed gene expression, as shown in the figure shown below, which we have now deleted from the manuscript. Lines 175-178 in original main text, described genetic regulation of gene expression, i.e. correlations between observed and predicted gene expression. Although correlation between imputed and observed gene expression may translate into correlation of phenotype trait associations between the 2, it is generally not since genetic regulation of gene expression only explained, on average, 12% and 7% of the observed gene expression variance as judged by the R2 values of the predictive models. We will make all the data used in figures available as supplements.

Reviewer 1, Comment 22. 444: “The non-coding CF modifier genes are likely underestimated” - in addition, they seem understudied from the fewer references in table 1 compared to the protein coding genes. Can you elaborate on the reasons why this may be?

Response: According to one review (PMID:28815535) (13), functional relevance of non-coding RNAs in general received little attention by the scientific community in the pre-genome sequence era before 2000s. Publication of the reference human genome sequence in 2001 highlighted the small portion of the genome sequences that code for proteins, and the advancement of deep sequencing of RNA in recent years allowed the discovery and cataloging of many transcribed genome sequences into non-coding RNAs. These RNAs were only identified in recent years, and there are relatively few reagents, such as enzyme assay and antibodies used to study proteins, to study non-coding RNAs in normal biology and disease associations.

The relevant text reads (lines 455-458): “The non-coding CF modifier genes reported here are likely under-estimated compared to protein-coding genes, due to reference genome and gene annotations associated with some of the gene expression data sets used in predictive model training, and general lag of functional knowledge of non-coding transcripts (13).” 

Reviewer 1, Comment 23. 460-461: “Furthermore, the number of genes whose expression can be reliably predicted from genetic variants varied among tissues” - a noticeable absence throughout most of this paper is the lack of individual tissues being scrutinized. Which tissue are these significant genes in, and how may that also contribute to CF? And if the tissue seems irrelevant, why are the findings still important?

Response: We will provide association testing results from each tissue, and the 2 meta-analyses results combining all tissues by p-values. As discussed above, all the tissues may contribute to generate predictive models of gene expression by cis-SNPs since majority of such regulation are preserved across tissues. The importance of genetic regulation of gene expression to CF lung disease is linked to the fact that SNPs associated with the disease phenotype by GWAS are part of the predictive models of some genes. While some genetic diseases have direct tissue origins, e.g., sickle cell anemia in red-blood cells, CF has a complex disease pathogenesis involving a chain of events in multiple tissues, interactions with the environment (reduced clearance of pathogens), and great variations in disease manifestations even among patients with the same genetic defects in CFTR. The time and place genetic regulation of modifier genes of CF diseases are active and meaningful, in terms of affecting disease outcome, are not clear. We therefore are trying to use all the available data at present to generate leads and hypothesis. 

Reviewer 1, General Comment: “Next, review figures/tables and supplementals” 

Reviewer 1, Comment 24. Table 1: add a column of which tissue the P-value was max. Where did the keywords originate from? Can you use coefficients instead of protective/harmful?

Response: The maximal p-value was retrieved among 4 meta-analysis p-values, each from various number of tissues depending on specific genes. With the limited space in Table 1 format, it is difficult to provide which tissues contributed to the p-value displayed, but the information will be provided in supporting files. The functional keywords were added by us, through review of the literature and our current expert knowledge of CF disease pathogenesis. It is subjective, and not from any public annotation database. We have added beta coefficient from linear regression of PrediXcan imputed gene expression to CF lung disease phenotype, as well as TWAS signed zscore, both represent mean estimated effect sizes among the tissue models tested. 

Reviewer 1, Comment 25. Fig. 1: Since the PM, imGE, and pvals for GTEx, DGN, nasal epi, and LCL are similar except for sample sizes, can they be consolidated to look less cluttered? Also, since genotypes were used to make the gene expression models for nasal epi and LCL and not summary statistics, “GWAS imputation” may be misleading.

Response: We have simplified the workflow into 2 arms – PrediXcan imputed gene expression, and TWAS off summary statistics. To clarify, the starting data of GWAS imputation represent common SNP dosages imputed for each individual patient, which are required for predictive model training. The GWAS (SNP) imputation was analyzed as traditional GWAS according to the same protocol outlined in the Corvol, et al, paper (3), and the summary results were used as input for TWAS/FUSION analysis.

Reviewer 1, Comment 26. Fig. 2: There’s a lot going on in this plot that makes it difficult to follow. Is it more legible if you subset to only the most important tissues?

Response: As expanded upon elsewhere in this document, we feel it is relevant to show results from all the tissues, since the different tissues are means to deduce expression regulation by genetic variance, and many signals were derived from “non-CF” tissues. However, we understand that there is a lot going on in the plot, and we have made an attempt within the text to further highlight the major patterns in the figure to help the reader navigate the figure (lines: 225-232). These major features include a general agreement between PrediXcan and TWAS, and the direction of imputed expression change to CF lung disease phenotype are consistent among different tissues in general. Additionally, the strongest signals are near GWAS loci on chr5 and chr6, which are imputed in most tissues.

Reviewer 1, Comment 27. Fig. 3: Use ggrepel with higher force for easier readability. These plots would be more appropriate in the supplemental figures with the r2 of both of them described in the text.

Response: We appreciate the pointer, have tried to improve the readability and have moved the figure to the supplements as the reviewer suggested.

Reviewer 1, Comment 28. Fig. 4: At a glance, the PrediXcan and TWAS results look similar in architecture. Can these results be consolidated with colors identifying differences between the methods instead, or colors indicating known vs novel CF genes?

Response: Thank you for the suggestion! We have now consolidated the Manhattan plots to show PrediXcan, TWAS, and GWAS results in the same figure (which is now Figure 3 in the main text). For the PrediXcan (A) and TWAS (B) results, the red squares represent genes near GWAS loci, and the blue triangles represent novel genes outside of 1 MB from the GWAS loci.

Reviewer 1, Comment 29. Fig. 5: This figure would also benefit from ggrepel. It would be easier to follow as a 2x2 rather than branching off the original scatterplot. Are all 3 of these offshoots necessary? What do each of them represent that the others don’t?

Response: Based on this comment, we have simplified the figure (now Figure 4 in the main text) to represent just 1 scatterplot without gene name labels. The gene level information will be available in table format. The colored markers represent the following: red squared = consensus modifier genes near GWAS loci; blue triangles = consensus genes outside GWAS loci; and the black diamonds = genes near GWAS loci, that are not supported by gene expression imputation to be associated to CF lung disease. The last category represents the usefulness of data mining in eliminating certain genes near GWAS loci due to lack of eQTL support from currently available data sets. 

Reviewer 1, Comment 30. Fig. 6: Very unique and insightful visual. Can you discuss more about how these genes are co-regulated from their intertwined, linked eQTLs?

Response: Yes, to paraphrase the relationship between GWAS, eQTL, and gene expression imputation: eQTLs from independent training data sets help select SNPs that are correlated with gene expression changes, and if the selected SNPs are associated with the phenotype of interest from GWAS collectively, then the imputed gene expression from these SNPs will be associated with the phenotype. Since genes near a particular GWAS loci may be co-regulated (or correlated with expression, eQTL) by the same variant(s), they would also be associated with the phenotype through gene expression imputation, as shown for chr3, chr5, and chr6 (supporting figures S8-S10). 

Reviewer 1, Comment 31. Fig. 7: Put the r2 in captions. This figure would also be better suited for a supplemental.

Response: Based on the previous comment (Reviewer 1, Comment 16), we have decided to remove this figure from the manuscript.

Reviewer 1, Comment 32. Fig. 8: Can you differentiate the known genes from the novel ones with color, bolding, or other formatting? Do you have similar figures for “consensus” genes available?

Response: We have had to update the results of gene set and pathway enrichment analyses since we have updated the PrediXcan association analysis to utilize 4 genotype PCs (instead of 1 in the original report) to control for population stratification (as requested by Reviewer 2), but the original feedback regarding the figure is still relevant. To clarify, the Gene Set Enrichment Analysis (GSEA) interrogates the rankings of genes of particular set or pathway among all imputed genes, we are leveraging concerted changes in ranks from both strong and weak signals. As a result, an enriched pathway may not contain top consensus modifier genes (although many of them do, e.g. HLA containing gene sets). We have added color highlights to the gene hash marks as follows: red = candidate modifiers (379) near GWAS loci; blue = candidate modifiers outside GWAS loci.

Reviewer 1, Comment 33. Fig. 9: Color in the HLA gene dots, they get lost amongst all the grey.

Response: We have marked the HLA genes with red triangles to distinguish them from other genes. Note: this figure is now Figure 7A.

Reviewer 1, Comment 34. Supporting methods: I enjoy your well-described and well-cited (with links!) descriptions. There are a few very minor typos - “the imputed gene expression data sets had sample size” should be “the imputed gene expression data sets had a sample size”. 

Response. Thank you. We have significantly revised the Supporting methods document, the exact phrase is no longer there. .

Reviewer 1, Comment 35. I have a small concern with the hierarchical clustering analysis. Why did you have “the missing values in the resultant distance matrix were replaced with the largest distance values between any pairs” rather than just leaving the data N/A? It seems misleading. Also, again, for a PrediXcan-like method that also uses summary statistics, S-PrediXcan is up your alley.

Response: Unfortunately, the current clustering function hclust from R does not tolerate missing values (although distance calculations works fine with NA for missing values), we therefore chose the largest distance, analogous to setting a threshold to treat the missing values as the most dissimilar pairs for clustering purposes. Note: this only affects the clustering tree (which seems within reasonable shape in Figure 2), and the heatmap still uses the values and color NA as white. As described above, we did explore S-PrediXcan and obtained similar, but not identical, results from PrediXcan, and decided to stay with the current format, since overall statistical power is better with PrediXcan. 

Reviewer 1, Comment 36. S1 Fig.: Do you think these dramatically different slopes are due to RNA quantification collection methods, sample sizes, or other outside factors?

Response: Mainly sample size, since the x-axis is simulated null distribution of r2, which depends on sample size, and the sample sizes for nasal epithelial biopsies and LCLs are 132 and 753.

Reviewer 1, Comment 37. S2 Fig.: I quite enjoy this figure. Can you compare the LCL microarray observed/expected gene count to those found in PrediXcan microarray-based models?

Response: We did not find microarray based models from PrediXcan, but a similar observation was reported by TWAS (9) (Figure 3 from their paper – see below), where NTR (the Netherlands Twins Registry) data set from microarray had lower slope between gene count versus sample size (green in the reproduced figure).

Reviewer 1, Comment 38. S6 Fig.: This is illegible. Would you be able to give each point a number and then have a side table with both the numbers and gene names?

Response: It is not feasible to label all the 52 consensus genes, so we chose to label just the top few with p-value<10-05. The full data (non-graphed) will be provided in supporting tables.

Reviewer 1, Comment 39. S11 Fig.: Can you include within the main text as part of the discussion why CFTR isn’t as highly significant as one would initially think in an analysis of cystic fibrosis?

Response: Yes, we are testing severity of lung disease among CF patients, many of them have the same CFTR mutations, F508del homozygotes being the most common. Because a CFTR deleterious mutation is a criterion for inclusion, we are unlikely to see further association at the locus. 

Reviewer 1, Comment 40. S Tables: Can you bold or italicize the known or novel gene findings to make them easier to differentiate?

Response: We have now highlighted the consensus genes in the supporting tables as suggested by the reviewer. 

Reviewer 1, Final Comment. Overall, my concerns lie mainly with the comparison between PrediXcan and TWAS, the interpretability of the figures, and the lack of connecting the actual tissue genes were determined significant to the CF phenotype, but I enjoy your analysis and your contribution to determining the genetic architecture of CF. Addressing these concerns as well as clarifying the points I made above and adding an additional colocalization analysis would strengthen this paper.

Response: We thank the reviewer for the very detailed reading of the manuscript and appreciate the time and thoughtful feedback! Hopefully, we have addressed the reviewer’s concerns.

Reviewer 2, General comment: I have two questions:

Reviewer 2, Comment 1. 1. The harmonic mean P-value method was recently corrected ( please see: http://blog.danielwilson.me.uk/2019/08/updated-correction-harmonic-mean-p.html ). Do the authors' calculations incorporate the updated (and corrected) harmonic mean P-value method?

Response: We thank the reviewer for alerting us to the method update, and as a result, we have re-run the relevant analyses. Apparently, we were already using version 3 as the re-run results were identical to before. 

Reviewer 2, Comment 2. 2. There is no discussion of LD-contamination ( see for example: https://www.nature.com/articles/s41467-018-03621-1 ). To what extent does this affect the results?

Response: LD-contamination refers to SNPs in strong LD, among which one or more are projected to be causal. Although LD-contamination is a major challenge in determining causal variant in an explanatory statistical model due to collinearity of SNPs in LD, it does not affect the accuracy of predictive models, which was used in the context of predicting gene expression from SNPs, refer to https://www.theanalysisfactor.com/differences-in-model-building-explanatory-and-predictive-models/, and https://statisticalhorizons.com/prediction-vs-causation-in-regression-analysis. More in depth discussion on the topic, see https://www.stat.berkeley.edu/~aldous/157/Papers/shmueli.pdf. We believe LD-contamination does not affect the accuracy of gene expression prediction from SNPs in LD, therefore the results of our data mining at gene level was not affected. It does present great challenge to identify causality at both SNP and gene levels, which require further experimental studies.

Reviewer 3.

Reviewer 3, Overall Comments: On the whole, the writing of the manuscript can and should be improved. As it stands, it is hard to follow. The authors present no motivation for the selected approaches to analyze the data (and why they chose to use two). The methods are not described well. It takes a great effort to match the description of the analyses with Fig. 1 illustrating them.

Response: Hopefully, we have improved our manuscript through the revision and resolved the issues raised by the reviewers.

Reviewer 3, Comment 1. Description of one type of analysis is interweaved with sentences starting with ‘Alternatively’ (lines 120, 132) describing the other type of analysis. This makes it harder to follow either of them.

Response: We understand this to be a general criticism that perhaps we had moved too many details of our analyses to the Supporting Methods.We have tried to strike a new balance, and re-organized the section to describe the 2 different approaches separately.

Reviewer 3, Comment 2. The authors pay attention to unimportant details (such as talk about GTEx pilot data) but not discuss important ones (e.g. choice of just 1 principal component to correct for ancestry).

Response: We thank the reviewer for the constructive comment. We will explain our reason to discuss GTEx pilot data in response to Comment 13.

Regarding the use of 1 PC, since our patent demographics are >90% Caucasian, we originally chose just 1 PC as a covariate to control for population stratification. However, after careful consideration, we agree with the reviewer that this was inadequate, and we now utilize 4 PCs as covariates in association testing. This did reduce the number of consensus and potential candidate modifier genes to 52 and 379 (from 54 and 531 when we were using 1 PC). This approach is more conservative and is based upon variance explained (Supporting Figure S4). We believe it to be the best decision. The results were updated in the current revision.

Reviewer 3, Comment 3. It causes great concern to see that the authors did not correctly use scientific E-notation for small numbers.

 For instance, instead of ‘1e-6’ or ‘10^{-6}’ the authors have 1x10e-06, which would actually be equal to 1e-5 if read correctly (10x1e-6 = 1e-5).

 There are a total of seven instances of incorrect use of scientific E-notation.

Response: We are very sorry for our carelessness, and we thank the reviewer for pointing out our mix-up. We have corrected all instances. 

Reviewer 3, Comment 4. Which samples were whole genome sequenced (WGS)?

Response: None used in this analysis. Gene expression from nasal epithelial samples were obtained from RNA-seq and LCLs from microarray (GSE60690). The 101 Canadian WGS samples mentioned were used to form a reference population panel for GWAS imputation of the array genotyped samples (5), but it was not directly used in this analysis. We have reviewed the method sections and added clarifying language where we thought it would help.

Reviewer 3, Comment 5. Is the sequencing data publicly available?

Response: The RNA-seq from nasal epithelial samples are not currently available. We have initiated the process to deposit the raw RNA-seq data into controlled access database, dbGaP in compliance with patient consent and IRB.

Reviewer 3, Comment 6. Were these samples among those being imputed?

Response: Yes, all samples were used for imputation, but they were excluded in association testing if they contributed to model building.

Reviewer 3, Comment 7. Were WGS genotypes mixed with imputed genotypes in the analyses?

Response: No. The 101 Canadian WGS samples were used to combine with 1000 genome projects phase3 (v5a) haplotype data to form a hybrid reference for GWAS imputation (5), and they were not used in this analysis due to data provenance, since the WGS sample are not part of GWAS cohorts.

Reviewer 3, Comment 8. Line 127. Why only one PC was used for correction of ancestry? This appears to be grossly insufficient given current knowledge.

Response: After more careful consideration with the reviewer feedback, we updated the analysis by including 4 PCs (Supporting Figure S4).

Reviewer 3, Comment 9. Line 129. There are numerous methods for robust regression analysis. Not providing a name for the chosen method, only a citation is a great inconvenience for the reader.

Response: We had provided more details in the Supporting methods in our original manuscript, but we have now added additional information in the main text in the revision as suggested by the reviewer. Revised text (142-143): “…the robust regression utilized iterated re-weighted least squares by the rlm function from the R package, MASS.” 

Reviewer 3, Comment 10. Lines 137-138. What does “in our most stringent analyses” mean?

Response: We meant consensus, which is hopefully explained above in the overview and response to Reviewer 1. We have revised the text (lines 160-164): “For significant modifier genes from each analysis platform, a p-value < 0.01 from both the HMP, and correlation adjusted method (EBM for PrediXcan, or omnibus for TWAS) was chosen. Consensus between the 2 result sets (with 4 p-value < 0.01 thresholds) yielded the most robust findings, while the union of significant genes from 2 result sets maximized sensitivity of discovery.”

Reviewer 3, Comment 11. Line 138. The phrase “we sought consensus” does not exactly read as “we selected genes significant in both analyses”.

Response: Noted. We have tried to improve the precision of our descriptions. The revised text (lines 157-164): “Multi-tissue tests from each result set were combined by two separate meta-analysis methods, a simple harmonic mean p-value (HMP) (19), and a correlation adjusted method, specifically, empirical adaptation of Brown’s method (EBM) (20) for PrediXcan, or omnibus test (10) for TWAS. For significant modifier genes from each analysis platform, a p-value < 0.01 from both the HMP, and correlation adjusted method (EBM for PrediXcan, or omnibus for TWAS) was chosen. Consensus between the 2 result sets (with 4 p-value < 0.01 thresholds) yielded the most robust findings, while the union of significant genes from 2 result sets maximized sensitivity of discovery.”

Reviewer 3, Comment 12. Line 173. What is the definition of 'imputable gene'?

Response: It is defined as genes that showed significant component of genetic regulation defined by PrediXcan and TWAS predictive model builders – for PredictDB models based on GTEx v7 data, the cross-validated r>0.1 and p-value<0.05 were used, which are the same for our CF derived models as well; and for TWAS, significant h2 estimates was used (http://gusevlab.org/projects/fusion/). 

Reviewer 3, Minor Comments

Reviewer 3, Comment 13. Line 72. Why even mention GTEx pilot data?

Response: Our strategy was designed to consider the general characteristics of genetic regulation of gene expression, or eQTL. These were initially described in publications by GTEx using the pilot data and were not fully reiterated upon later data release. 

Reviewer 3, Comment 14. Why every plot is black and white?

Response: Thank you. We have altered the figures to incorporate color where helpful.

 

References

1. Consortium GT, Laboratory DA, Coordinating Center -Analysis Working G, Statistical Methods groups-Analysis Working G, Enhancing Gg, Fund NIHC, et al. Genetic effects on gene expression across human tissues. Nature. 2017;550(7675):204-13.

2. Gamazon ER, Segre AV, van de Bunt M, Wen X, Xi HS, Hormozdiari F, et al. Using an atlas of gene regulation across 44 human tissues to inform complex disease- and trait-associated variation. Nat Genet. 2018;50(7):956-67.

3. Corvol H, Blackman SM, Boelle PY, Gallins PJ, Pace RG, Stonebraker JR, et al. Genome-wide association meta-analysis identifies five modifier loci of lung disease severity in cystic fibrosis. Nat Commun. 2015;6:8382.

4. Taylor C, Commander CW, Collaco JM, Strug LJ, Li W, Wright FA, et al. A novel lung disease phenotype adjusted for mortality attrition for cystic fibrosis genetic modifier studies. Pediatr Pulmonol. 2011;46(9):857-69.

5. Panjwani N, Xiao B, Xu L, Gong J, Keenan K, Lin F, et al. Improving imputation in disease-relevant regions: lessons from cystic fibrosis. NPJ Genom Med. 2018;3:8.

6. Gamazon ER, Wheeler HE, Shah KP, Mozaffari SV, Aquino-Michaels K, Carroll RJ, et al. A gene-based association method for mapping traits using reference transcriptome data. Nat Genet. 2015;47(9):1091-8.

7. Marazzi A, Joss J, Randriamiharisoa A. Algorithms, routines, and S functions for robust statistics : the FORTRAN library ROBETH with an interface to S-PLUS. Pacific Grove, Calif.: Wadsworth & Brooks/Cole Advanced Books & Software; 1993. xii, 436 p. p.

8. Venables WN, Ripley BD, Venables WN. Modern applied statistics with S. 4th ed. New York: Springer; 2002. xi, 495 p. p.

9. Gusev A, Ko A, Shi H, Bhatia G, Chung W, Penninx BW, et al. Integrative approaches for large-scale transcriptome-wide association studies. Nat Genet. 2016;48(3):245-52.

10. Yang J, Bakshi A, Zhu Z, Hemani G, Vinkhuyzen AA, Lee SH, et al. Genetic variance estimation with imputed variants finds negligible missing heritability for human height and body mass index. Nat Genet. 2015;47(10):1114-20.

11. Yang J, Benyamin B, McEvoy BP, Gordon S, Henders AK, Nyholt DR, et al. Common SNPs explain a large proportion of the heritability for human height. Nat Genet. 2010;42(7):565-9.

12. Wainberg M, Sinnott-Armstrong N, Mancuso N, Barbeira AN, Knowles DA, Golan D, et al. Opportunities and challenges for transcriptome-wide association studies. Nat Genet. 2019;51(4):592-9.

13. Jarroux J, Morillon A, Pinskaya M. History, Discovery, and Classification of lncRNAs. Adv Exp Med Biol. 2017;1008:1-46.

---

## [Editor Report · Decision Letter 1]

2 Sep 2020

Mining GWAS and eQTL data for CF lung disease modifiers by gene expression imputation

PONE-D-19-23859R1

Dear Dr. Dang,

We’re pleased to inform you that your manuscript has been judged scientifically suitable for publication and will be formally accepted for publication once it meets all outstanding technical requirements.

Kind regards,

Dylan Glubb

Academic Editor

PLOS ONE

Additional Editor Comments (optional):

I congratulate the authors for their considered and well explained responses to the reviewers which were a pleasure to read. However, I would suggest that it is unlikely that transcription factors would provide targets for intervention (see lines 438-9). If the authors are interested in finding druggable targets amongst their candidate genes, I would recommend using the Open Targets database (https://www.targetvalidation.org/) to assess this.
---

## [Editor Report · Acceptance letter]

9 Nov 2020

PONE-D-19-23859R1 

Mining GWAS and eQTL data for CF lung disease modifiers by gene expression imputation 

Dear Dr. Dang:

I'm pleased to inform you that your manuscript has been deemed suitable for publication in PLOS ONE. Congratulations! Your manuscript is now with our production department. 

Kind regards, 

on behalf of

Dr. Dylan Glubb 

Academic Editor

PLOS ONE